

# Validation of ozone profile retrievals derived from the OMPS LP version 2.5 algorithm against correlative satellite measurements

**Natalya A. Kramarova[1,2], Pawan K. Bhartia[2], Glen Jaross[2], Leslie Moy[1], Philippe Xu[3], Zhong Chen[1], Matthew DeLand[1], Lucien Froidevaux[4], Nathaniel Livesey[4], Douglas Degenstein[5], Adam Bourassa[5], Kaley A. Walker[6], and Patrick Sheese [6]**

[1]Science Systems and Applications, Inc., Lanham, Maryland, USA
[2]NASA Goddard Space Flight Center, Greenbelt, Maryland, USA
[3]Science Applications International Corp., Beltsville, Maryland, USA
[4]Jet Propulsion Laboratory, California Institute of Technology, Pasadena, California, USA
[5]University of Saskatchewan, Saskatoon, Canada
[6]University of Toronto, Department of Physics, Toronto, Canada

*Correspondence to*: Natalya A. Kramarova (natalya.a.kramarova@nasa.gov)

**Abstract.** The Limb Profiler (LP) is a part of the Ozone Mapping and Profiler Suite launched on board of the Suomi NPP satellite in October 2011. The LP measures solar radiation scattered from the atmospheric limb in ultraviolet and visible spectral ranges between the surface and 80 km. These measurements of scattered solar radiances allow for the retrieval of ozone profiles from cloud tops up to 55 km. The LP started operational observations in April 2012. In this study we evaluate more than 5.5 years of ozone profile measurements from the OMPS LP processed with the new NASA GSFC version 2.5 retrieval algorithm. We provide a brief description of the key changes that had been implemented in this new algorithm, including a pointing correction, new cloud height detection, explicit aerosol correction, and a reduction of the number of wavelengths used in the retrievals. The OMPS LP ozone retrievals have been compared with independent satellite profile measurements obtained from the Aura Microwave Limb Sounder (MLS), Atmospheric Chemistry Experiment Fourier Transform Spectrometer (ACE-FTS) and Odin Optical Spectrograph and InfraRed Imaging System (OSIRIS). We document observed biases, seasonal differences and evaluate the stability of the version 2.5 ozone record over 5.5 years. Our analysis indicates that the mean differences between LP and correlative measurements are well within required ±10% between 18 and 42 km. In the upper stratosphere and lower mesosphere (>43 km) LP tends to have a negative bias. We find larger biases in the lower stratosphere and upper troposphere, but LP ozone retrievals has significantly improved in version 2.5 compared to version 2 due to the implemented aerosol correction. In the northern high latitudes we observe larger biases between 20 and 32 km due to the remaining thermal sensitivity issue. Our analysis shows that LP ozone retrievals agree well with the correlative satellite observations in characterizing vertical, spatial and temporal ozone distribution associated with natural processes, like the seasonal cycle and Quasi Biennial Oscillations. We found a small positive drift ~0.5%/yr in the LP ozone record against MLS and OSIRIS that is more pronounced at altitudes above 35 km. This pattern in the relative drift is consistent with a possible 100-meter drift in the LP sensor pointing detected by one of our altitude resolving methods.



# 1 Introduction

Since late 1980s the production of the human-made halogen compounds that destroy the stratospheric ozone layer has been strictly regulated. Observations show that the concentration of ozone-destroying gases in the atmosphere is declining (WMO, 2014), and stratospheric ozone is expected to recover to the 1980 level over the next several decades. However, the detection

of the stratospheric ozone recovery is complicated by a competing effect from increasing concentration of greenhouse gases (that also lead to rises in stratospheric ozone due to stratospheric cooling) and large uncertainties in the measurements (e.g. Harris et al., 2014). Unexpected variations in the atmospheric circulation, not captured by models, such as the recent disruption of the quasi-biennial oscillation (Newman et al., 2016, Tweedy et al., 2017), contribute to additional noise in the trend estimates and emphasize the continued importance of high quality ozone measurements.

The Ozone Mapping and Profiler Suite (OMPS) represents a new generation of the U.S. ozone monitoring sensors (Flynn et al., 2006). Suomi NPP OMPS serves as a bridge mission connecting Backscatter Ultraviolet (BUV) global ozone measurements pioneered in the 1970s with the next-generation of NASA/NOAA sensors on board the Joint Polar Satellite System (JPSS). OMPS was designed to provide profile and total ozone measurements to extend the long-term historical ozone record into the future to monitor the atmospheric ozone recovery.

The OMPS consists of three ozone-acquiring sensors. All three sensors measure scattered solar radiances in overlapping spectral ranges and scan the same air masses within 10 minutes. The nadir module of the OMPS combines two sensors that share some optical elements: the Total Column Nadir Mapper (TC-NM) for measuring total column ozone, and the Nadir Profiler (NP) for ozone vertical profiles. The Limb Profiler (LP) module is designed to measure vertical ozone profiles with high vertical resolution (~2-3 km) from the upper troposphere to the mesosphere.

The nadir OMPS sensors are based on heritage designs of Total Ozone Mapping Spectrometer (TOMS) and Ozone Mapping Instrument (OMI) for OMPS TC-NM and Solar Backscatter Ultraviolet (SBUV/2) for OMPS NP. OMPS LP sensor development became possible after technological investments NASA made in the 1990s by flying the Shuttle Ozone Limb Sounding Experiment on two Space Shuttle missions STS-87 and STS-107 (Flittner et al., 2000; McPeters et al., 2000) to demonstrate the viability of the limb scattering technique to produce a high vertical resolution ozone profile record

comparable in quality to more expensive limb-viewing thermal infrared and microwave sensors. The next Limb Profiler is scheduled for a launch in 2022 on board of the Joint Polar Satellite System-2 (JPSS-2) satellite. The OMPS LP is aimed to continue the high vertical resolution ozone observations from NASA's solar occultation (Stratospheric Aerosol and Gas Experiment (SAGE), Halogen Occultation Experiment (HALOE)) and limb emission missions (MLS). The limb scattering technique, employed in the OMPS LP, has been already successfully used for retrieving atmospheric ozone and aerosol

profiles in several satellite missions, such as SAGE III (Rault, 2005; Rault and Taha, 2007), Odin OSIRIS (Llewellyn at al., 2004), Scanning Imaging Absorption Spectrometer for Atmospheric Cartography (SCIAMACHY) (Bovensmann et al., 1999), and Global Ozone Monitoring by Occultation of Stars (GOMOS) (Tukiainen et al., 2011; Taha et al., 2008).

In summer 2017 all OMPS LP measurements starting from April 2012 have been processed with the new version 2.5





algorithm. This study documents the changes implemented into the OMPS LP ozone retrieval algorithm and thoroughly evaluates the new version 2.5 ozone profile dataset. The assessment of the stability and systematic uncertainties of the new OMPS LP product and estimation of effects of calibration updates and algorithmic changes on the ozone product are crucial. The goals of this validation are: a) to isolate errors in ozone related to instrument or algorithmic effects from real

geophysical signals; and b) to estimate a total error budget. To evaluate the performance of the new version 2.5 ozone profile dataset we compare this dataset with the previous version 2 processing and with independent satellite profile measurements from three satellite instruments: Aura MLS, ACE-FTS and Odin OSIRIS.

In Section 2 we describe the algorithmic and instrument calibration changes that had been implemented for the version 2.5 processing. Correlative satellite ozone datasets and validation methods are described in Sec. 3. In Section 4 we describe

results of the validation that include internal analysis of measured radiances and examination of the OMPS LP systematic errors, as well as comparisons with independent measurements. Conclusions are presented in Sec. 5. The Supplement provides additional supporting information.

## 2. Version 2.5 ozone retrieval algorithm for the OMPS Limb Profiler

The first version of LP ozone retrievals was released soon after the beginning of operational observations in April 2012. The

original ozone retrieval algorithm for the OMPS LP is fully described by Rault and Loughman (2013). In July 2014 version 2 of the LP ozone profile dataset was released. Here we present a new version 2.5 of the LP ozone data that have been publically released in July 2017 (Kramarova et al., 2017). Both datasets (version 2 and version 2.5) have been produced using a modified version of the OMPS LP retrieval algorithm. LP ozone retrievals are reported daily in HDF files. These files contain ozone number density retrievals from the center slit only (see Sec. 4.1) along with geolocation information and

quality flags (DeLand, 2017). For users' convenience, for each ozone measurement we also report atmospheric pressure and temperature profiles that are derived from the MERRA-2 reanalysis system. Specifically, we use the Forward Processing-Instrument Team (FP-IT) product produced by the NASA Global Modeling Assimilation Office (GMAO) (Gelaro et al., 2017).

In this section we provide a brief description of key changes implemented in the retrieval algorithm for processing version 2

and version 2.5 datasets. Table 1 gives a summary of the algorithm modifications.

### 2.1 OMPS Limb Profiler

The OMPS LP sensor measures solar radiances scattered from the atmospheric limb in the ultraviolet (UV) and visible (VIS) spectral ranges. The UV measurements are used to retrieve ozone concentration in the upper and middle stratosphere, and visible measurements are used to retrieve ozone in the lower stratosphere (McPeters et al., 2000). To expand the sensor

cross-track coverage the LP has three slits separated horizontally by $4.25^0$ (about 250 km). With 14 orbits per day and roughly about 160-180 measurements per orbit (~$1^0$ latitude sampling) the LP provides full global coverage every 3-4 days (Kramarova et al., 2014).





Each LP slit has a $1.85^0$ vertical FOV that corresponds to 112 km vertical extent at the tangent point. This allows LP to cover an altitude range from 0-60 km plus an offset allowance. The Charge Coupled Device (CCD) detector used in the LP can simultaneously collect scattered solar radiances from all altitudes in the spectral range between 290 nm and 1000 nm. The spectral resolution of the LP measurements varies from about 1.5 nm around 290 nm to 40 nm for longer wavelengths near

1000 nm. In order to cover the entire spectral and vertical range the OMPS LP splits each vertical profile into two images: low altitude (bright) signals are measured with the small aperture, whereas the high altitude (low intensity) signals are measured with the large aperture. Measurements at short and long integration times are interleaved over the nominal 19-second measurement duration (Jaross et al, 2014). All six spectra (three slits and two apertures) are captured onto a single focal plane, and three ozone profiles (one for each slit) are produced every 19 seconds. Due to bandwidth limitations data

from all CCD pixels can not be downloaded from the instrument to the ground during the normal operations, and only a relatively small subset of pixels, called the Sample Table, becomes available for the retrieval process. Data from downloaded pixels go through an intermediate step, in which the raw measured radiances get remapped onto regular (spectral and vertical) grids. The vertical sampling of LP measurements is ~1 km, but the actual Instrumental Field of View (IFOV) is about 1.3-1.7 km.

**2.2 OMPS LP ozone retrieval algorithm adjustments**

The OMPS LP retrieval algorithm employs the Optimal Estimation method to retrieve ozone profiles (Rodgers, 2000; Rault and Loughman, 2013). The Gauss-Seidel limb scattering (GSLS) radiative transfer model is used (Herman et al, 1994; Loughman et al., 2005; Loughman et al., 2015) to simulate limb-scattered radiances. The Bass and Paur (Bass and Paur, 1985) ozone cross-sections are used in the LP algorithm. The a priori ozone profiles are based on the seasonal ozone

climatology derived from Aura MLS and ozone sonde observations (McPeters and Labow, 2012). The scene reflectivity is derived from the LP measurements at 675 nm.

The LP retrieval algorithm is designed to independently retrieve ozone profiles from UV and VIS measurements using wavelengths pairs in UV range and triplets in VIS (Flittner et al., 2000; Rault et al., 2005; Roth et al., 2007). To minimize the sensitivity to the underlying scene reflectance and absolute instrument calibration, measured radiances are first

normalized with radiances measured at high altitudes. In the version 1 algorithm, UV radiances were normalized at 68.5 km and VIS radiances at 45.5 km. In version 2, the UV normalization altitude was lowered to 65.5 km. In version 2.5 the normalization altitudes were shifted even lower to 55.5 km and 40.5 km for UV and VIS radiances, respectively, to reduce contamination effects from Stray Light (SL) and Polar Mesospheric Clouds (PMC) contaminations.

Another key feature of the OMPS LP retrieval algorithm is the use of wavelength pairs in UV and triplets in VIS to minimize

the effect of aerosol scattering. In the UV spectral range, shorter wavelengths with strong ozone absorption are paired together with a weak absorbing wavelength around 353-355 nm. In the Chappuis band, wavelength triplets are used with one stronger absorbing wavelength close to the band peak balanced by two weaker absorbing wavelengths on either side of the peak around 500-510 nm and 673-675 nm.





Ozone profiles are retrieved independently from UV and VIS spectral ranges, so that the measurement vector $y$ consists of UV pairs or VIS triplets. In version 1, all available UV wavelengths between 280 and 325 nm paired with 355 nm have been used to acquire ozone profiles in the middle and upper stratosphere with typically 36-50 UV pairs being used depending on the Sample Table. In version 2, wavelengths shorter than 289 nm and wavelengths in the OH band (306.5-311 nm) were

excluded to avoid errors associated with stray light and OH contamination, respectively. We decreased the number of wavelength pairs to 3 in version 2.5. The three selected UV wavelengths (302 nm, 312 nm and 322 nm) are sufficient to cover an altitude range between 28.5 km and 50.5 km. The algorithm uses radiances from all three UV wavelengths at 50.5 km, then cuts the 302 nm radiance below 43.5 km and 312 nm radiance below 37.5 km. The main rationale behind the decision of using fewer wavelengths was to simplify the inversion algorithm in order to help calibrate LP measured

radiances (LP Level 1 product) and isolate errors associated with height registration, stray light contamination or radiometric calibrations (see discussion in Section 4.1).

In version 1, the retrieval algorithm for the VIS range used all wavelengths between 520 and 655nm with ~35 triplets typically being used. In version 2, the spectral range was reduced to 549-633 nm leaving ~17 triplets. In the current version 2.5 algorithm, we use only one VIS triplet with the central wavelength at 600 nm to retrieve ozone profiles in the altitude

range between 12.5 km or the cloud top (whichever is lower) up to 35.5 km.

The Optimal Estimation method requires defining the a priori covariance $S_a$ and measurement error covariance $S_\epsilon$ matrices. In the original LP retrieval algorithm the a priori covariance matrix was defined as the diagonal matrix with 100% variability. To improve stability of the ozone retrievals a Tikhanov's regularization term was added into the optimal estimation solution (see eq. 1 in Rault and Loughman, (2013)).

In version 2.5, the Tikhanov's regularization term was removed, but instead the a priori covariance matrices $S_a$ have been set assuming 25% ozone variability above 20 km, 50% ozone variability below 16 km and 5 km inter-level correlation:

$$S_a(i, k) = w(i) \cdot x_a(i) \cdot w(k) \cdot x_a(k) \cdot e^{\frac{-|z(i)-z(k)|}{L}} \qquad (1)$$

where $i$ and $k$ are indices for altitude levels, $x_a$ is the a priori ozone profile which varies with season (12 months) and latitude (eighteen $10^0$ latitude bins), $L=5.0$ km is the characteristic correlation length, $z$ is the LP vertical grid (0.5, 1.5, … 50.5), and

$w$ is a set of weighting coefficients set to 0.25 (or 25%) above 20 km and 0.5 (or 50%) below 16 km with a linear transition in between. These weighting coefficients provide a first order estimate for the vertical pattern of the natural ozone variability based on analysis of McPeters and Labow (2012) climatology.

The measurement noise covariance matrix $S_\epsilon$ was set as a diagonal matrix with each diagonal element being associated with the Signal to Noise Ratio (SNR) for the corresponding wavelength at a given altitude. The analysis of the random errors in

LP measurements showed that the SNR (which varies from 320 for 290 nm at 60 km to 1200 for 600 nm at 15 km) significantly underestimates the actual measurement noise (Jaross et al., 2014) that varies in a range between 0.5% and 1%. For version 2, the measurement noise was assumed to be ~1% for both UV and VIS retrievals. In the current version 2.5, the measurement noise is prescribed to be 1% in UV spectral range and 0.5% in VIS. According to Rodgers (2000), both $S_a$ and



$S_\epsilon$ covariance matrices can be considered as the "tuning parameters" for the retrieval algorithm to obtain reasonable retrievals. The main objective for our selection of these tuning parameters was to obtain a vertical resolution for ozone retrievals of ~ 1.7 km – 2.0 km across all altitudes which corresponds to the actual instrumental field of view for the LP sensor.

### 2.3 Cloud detection and aerosol retrievals

Clouds that are present along the line of sight of the LP sensor shield the radiances scattered from lower altitudes and reflect a large portion of the incoming radiation, which makes it extremely difficult to simulate limb scattered radiances and retrieve ozone values. For this reason the original OMPS LP algorithm was designed to identify a cloud and start ozone retrievals from 1 km above the cloud top height (Rault, 2005). A new approach for detecting cloud height was recently developed (Chen et al., 2016) and implemented in version 2.5. This approach uses the spectral dependence of the vertical gradient in radiance between two wavelengths in the visible and near-IR spectral bands to better discriminate between clouds and aerosols compared to earlier results when the radiance gradient from a single wavelength was used. Good agreement was found between LP retrieved cloud heights and coincident measurements by the Cloud-Aerosol Lidar and Infrared Pathfinder Satellite Observation (CALIPSO) sensor with the differences mostly range between -1 and +4 km and the median difference ~1.8 km (Chen et al., 2016).

The aerosol attenuation affects ozone retrievals even though the use of wavelengths triplets aimed to reduce those effects. However, in cases where the aerosol concentration in the stratosphere is high enough (after volcanic eruptions) or/and the ozone concentration is very low (e.g. tropical UTLS), an explicit aerosol correction is required for accurate ozone retrievals. The original OMPS LP retrieval algorithm had an aerosol correction module built into it. The aerosol profiles were retrieved first and then those profiles were used in the ozone retrievals (Rault and Loughman, 2013). However, analyses revealed a problem with the aerosol correction module that produced artificial structures in the ozone profiles. In version 2 the aerosol correction module was completely turned off. The analysis of version 2 ozone retrievals demonstrated that the aerosol effects were largest in the tropical UTLS, causing significant, consistent biases against independent ozone observations. Recently the new aerosol retrieval algorithm for the OMPS LP has been developed (Loughman et al., 2017), and version 1 of the LP aerosol dataset was released in February 2017. These LP aerosol profiles are now used to explicitly correct for aerosol contamination in the version 2.5 ozone retrieval algorithm.

### 2.4 OMPS LP altitude registration

Accurate altitude registration is a key component of the ozone uncertainty budget because an error of just 100 m in tangent height (TH) can result in as much as 3% error in ozone at altitudes above 35 km. Shortly after the mission began, the OMPS team discovered that the pointing information provided by the spacecraft star tracker system had a significant offset (~ 1.4 km, later confirmed by other SNPP sensors, e.g. VIIRS), most likely due to a shift in the relative position of LP and the star




tracker system during launch. To complicate this matter, the star tracker system is not mounted close to OMPS, leading to additional latitude and seasonally dependent errors in sensor pointing at the OMPS location due to thermal flexure of the spacecraft bus. In addition, an unexpected thermal sensitivity of the instrument itself was discovered, causing vertical and spectral shifts in the data in the northern hemisphere (Jaross et al., 2014). Our analysis has indicated ~500 m shifts in vertical

registration in the northern hemisphere due to heating of the instrument, which causes it to warp and changes the optical path inside the instrument.

A large fraction of our effort in the past 5 years has been invested in development and implementation of two methods based on analysis of LP measured radiances (Moy et al, 2017) to resolve altitude registration independent of the star tracker. These methods are the Rayleigh scattering attitude sensing (RSAS) and absolute radiance residual method (ARRM). The RSAS

method determines absolute altitude errors using a ratio of radiances at 350 nm between 20 and 30 km, but, because the method is susceptible to aerosol interference, it is limited to latitudes and time periods with minimal aerosol contamination. The ARRM method looks at vertical gradients of short UV radiances at ~295 nm for altitudes between 40 and 65 km, and is suitable to monitor relative change in the altitude registration. The ARRM method can also detect absolute errors, but it is used here only to evaluate relative errors due to its sensitivity to instrument stray light and dependence on the quality of

pressure data at altitudes above 60 km. The combined accuracy of our two altitude registration methods is about ±200 m.

The RSAS results confirm a 1-1.5 km spacecraft initial pointing error, which is confirmed by another instrument on board the Suomi NPP satellite. In version 2 the static altitude corrections of 0.58 km, 1.18 km and 1.75 km for the left, center and right slit, respectively, had been implemented.

In the v2.5 processing, the static altitude corrections have been refined to 1.12/1.37/1.52 km (which is ~190 m change for the

center slit relative to version 2). Analysis of ARRM results revealed two distinct 100 m jumps in the LP pointing altitude, one from a known spacecraft pitch adjustment on 25 April 2013 and another caused by spacecraft inclination adjustment maneuvers near 5 September 2014. A +100 adjustment was therefore applied in version 2.5 on each of these two dates. The ARRM results were also used to create a slit based, intra-orbital, seasonally varying altitude corrections, shown on Figure 1, which have been employed in v2.5 as well. The intra-orbital correction varies by slit with the greatest intra-orbital change of

~400 m for the left slit, ~350 m for the center and ~300 m for the right. For all slits the greatest intra-orbital correction occurs between June and July. In the beginning of orbit corrections are positive ~+200 m and with close to 0 m corrections in the southern tropics. At the end of the orbit, in the northern hemisphere, corrections tend to be negative ~ -100 m.

## 2.5 Stray Light Correction

Jaross et al. (2014) provides a full description of the stray light (SL) corrections that had been implemented in the version 2

production. The SL corrections in both UV and VIS were based on the pre-launch point-spread-function (PSF) measurements. Jaross et al. (2014) also describe the thermal adjustments introduced to address an unexpected thermal sensitivity of the instrument.



Further analysis revealed that the PSF-based corrections are insufficient. In version 2.5, empirical corrections were employed for VIS measurements only to quantify the residual SL error. This empirical technique had been previously used to calibrate limb scattering measurements from GOMOS (Taha et al., 2008) and SAGE III (Rault, 2005). The technique derives an empirical scale factor (~1.2) that is used to multiply the existing PSF-based correction. The scaled PSF corrections have been

applied to the VIS measurements to reduce the residual SL. The SL corrections in UV are the same as in version 2.0 because this technique could not be applied in the UV.

## 3. Correlative satellite ozone datasets and validation methods

### 3.1 Aura MLS

The Microwave Limb Sounder (MLS) instrument on board the Aura satellite was launched on July 15, 2004 (Livesey et al.,

2006). MLS measures thermal microwave emission from the Earth's atmospheric limb to retrieve vertical profiles of atmospheric temperature and trace gases from the upper troposphere up to the mesosphere. The ozone profiles are derived from 240 GHz measurements and reported as volume mixing ratios on a fixed pressure scale. The MLS retrieval algorithm employs the standard Optimal Estimation technique (Rodgers, 2000). To account for the inhomogeneity along the line of sight, the MLS retrieval algorithm simultaneously retrieves several vertical profiles along the orbital track. The vertical

resolution of the MLS ozone retrievals in the stratosphere is ~ 3 km and increases to 5.5 km in the mesosphere.

In this study we use the newest version 4.2x of MLS data, and we filtered the data using recommendations provided by Livesey et al. (2017). Typical retrieval precisions and systematic errors for MLS ozone profiles are shown in Fig. 2. For validation purposes we limited MLS data to day-time observations around 2 pm local time and accepted data with solar zenith angles (SZA) smaller than 89 degrees. The previous version 3.3 of MLS ozone data was thoroughly characterized in

the recent assessment of the limb satellite measurements against ground based lidar and sonde observations (Hubert et al., 2016). Biases for MLS ozone measurements are typically about ±5% or less in the stratosphere with larger systematic biases in the tropical UTLS. Hubert et al. (2016) found that the Aura MLS ozone data record is stable with no significant drifts in the middle stratosphere.

### 3.2 ACE-FTS

The Atmospheric Chemistry Experiment - Fourier Transform Spectrometer (ACE-FTS) is a solar occultation limb sounder that operates in the 750 to 4400 cm$^{-1}$ spectral region. ACE-FTS was launched in August 2003 on board the SCISAT satellite and provides profiles of atmospheric temperature, pressure and more than 30 trace gases. Measurements are taken every 2 seconds twice per orbit, during sunset and sunrise. The ACE-FTS covers vertical range from ~5 km up to 150 km with the vertical Field of View (FOV) of ~3-4 km and a vertical sampling of ~2-6 km. Ozone profile retrievals are limited to altitudes

between 5 and 101 km. The retrieval algorithm employs the non-linear least-squares technique to fit the observed spectra to the simulated one using the forward model calculations (Boone et al., 2005).



We use the most recent version 3.6 of the ACE-FTS data (Boone et al., 2013) and apply filters to remove physically unrealistic outliers from the data (Sheese et al., 2015). Even after the screening, we found large anomalies for several ACE profiles in the tropical UTLS (<18 km) that led to significantly greater noise in the comparisons. We had to put additional screening removing all profiles for which ozone values between 14.5 and 17.5 km are 5 times larger or smaller than the ACE mean values for 20° S-20° N latitude. This screening removed about 20 profiles, which is less than 2% from all available ACE profiles in 20° S-20° N latitude bin.

Comparisons with MIPAS and MLS, presented by Sheese et al. (2016), showed that ACE-FTS ozone retrievals typically agree within 5% in the middle stratosphere and exhibit positive biases of ~10-20% in the upper stratosphere and lower mesosphere. ACE retrievals have the smallest precision in the middle stratosphere among the instruments considered in this study (see Fig. 2).

### 3.3 Odin OSIRIS

The Optical Spectrograph and Infrared Imager System (OSIRIS) instrument was launched in February 2001 on board the Odin satellite (Llewellyn et al., 2004). OSIRIS measures the limb scattered sunlight in the range from 280 nm to 800 nm with 1 nm spectral resolution in a manner very similar to OMPS LP. The instrument scans the atmospheric limb between 7 km and 70 km at a rate of 0.75 km/s and covers the latitude range between 82° S and 82° N with the ascending node equator crossing local time around 6 pm. The ozone profiles are retrieved using Multiplicative Algebraic Reconstruction Technique (Degenstein et al., 2009) between 10 km or cloud top and 60 km. The retrieval algorithm allows to simultaneously use and merge information from UV and VIS radiances. Retrieval precisions for OSIRIS are shown in Fig. 2.

As a limb scattering instrument, the OSIRIS is very sensitive to the accuracy of the sensor pointing altitude. Hubert et al. (2016) found a significant drift in OSIRIS ozone time series with increases versus altitude. Detailed analysis performed by Bourassa et al. (2017) showed that this drift in ozone time series is due to a systematic error in the sensor pointing altitude. The pointing drift has a seasonal component and increases in magnitude since 2010. The drift in the OSIRIS altitude registration is also responsible for the larger positive trends derived from the merged SAGE II / OSIRIS data set (e.g. Harris et al., 2015). In this study, we use the newest version 5.10 of OSIRIS ozone data with the corrected drift in the sensor pointing altitude (Bourassa et al., 2017). To validate OMPS LP ozone retrievals, we use OSIRIS data between April 2012 and June 2017.

### 3.4 Methodology

In order to compare ozone measurements derived from different instruments, it is important to consider the spatial and temporal coverage of the different instruments and properly set collocation criteria. Our spatial collocation criteria are the following: profiles should be within ±2° latitude from each other with the distance between them less than 1000 km. We require that observations have been made within ±12 hours for ACE/OMPS and OSIRIS/OMPS pairs. For instance, both



MLS and OMPS LP instruments are onboard polar orbiting satellites with the equator crossing time around 1:30 pm. Both instruments have a dense spatial and temporal sampling. For MLS/OMPS pairs the temporal criteria is limited to ±5 hours. Figure S1 in the Supplement shows ozone differences between OMPS LP and ACE-FTS as a function of the distance between the measurements. We found stronger dependence on distance in the lower stratosphere and also in the vicinity of

polar vortices, where an increase in distance might lead to sampling of different air masses with significantly different ozone morphology. Stricter coincidence criteria in these cases lead to a smaller spread for the differences (Fig S1 in the Supplement). However, our goal is to study global patterns in ozone fields, and a larger number of matching profiles gives more robust and statistically significant results. The collocation criteria chosen here provide a reasonable balance between the size of the statistical pool and the physical ozone variability.

In this study, we analyze ozone profiles on the LP native coordinate system (number density on altitude grid), which requires units conversion for MLS and ACE retrievals. MLS volume mixing ratios were first interpolated on the LP vertical altitude scale using MERRA-2 FP pressure profiles reported with OMPS LP data. These mixing ratios were converted into number densities using the MERRA-2 FP ancillary pressure and temperature profiles. We did not use MLS geopotential height profiles because of their known drift (Livesey et al., 2015; Hubert et al., 2016). ACE volume mixing ratios were converted to

number densities using retrieved ACE pressure values. Since ACE data are reported on the altitude scale, no interpolation was required. No unit conversion or vertical scale interpolation is needed for the OSIRIS data that are retrieved as number densities on the same altitude grid as OMPS LP.

If a cloud is detected, the LP algorithm does not retrieve ozone below the cloud top. We cut matching correlative ozone profiles at the cloud top altitude as well to avoid biases due to different sampling in cloudy and cloud-free conditions. In this

study, we do not account for small differences in the vertical resolution of the instruments, thus assuming that the vertical resolution is similar among considered instruments.

Using sets of matching profiles, we compute mean biases and the corresponding standard deviations to assess the accuracy and precision of the OMPS LP ozone profiles. The biases $b$ are calculated as an average difference between LP $\hat{X}_{OMPS}$ and reference $\hat{X}_{ref}$ instrument retrievals. The relative biases are calculated by dividing an average difference by the mean values

of the reference instrument:

$$b(z) = \frac{\sum_{n=1}^{N}(\hat{X}(z)_{OMPS} - \hat{X}(z)_{ref})}{N} \qquad (2)$$

where N is a number of matching profiles for each pair of instruments and $z$ is altitude.

The standard deviations for the biases are computed as:

$$\sigma(z)^2 = \frac{\sum_{n=1}^{N}(\hat{X}(z)_{OMPS} - \hat{X}(z)_{ref} - b(z))^2}{N} \qquad (3)$$

To investigate latitudinal patterns of the relative differences between pairs of instruments we calculated biases against all three correlative instruments in five wide (30 or 40 degree) latitude bins using all matching profiles that fall in each latitude bin (90°S - 60°S, 60°S - 20°S, 20°S - 20°N, 20°N-60°N, and 60°N-90°N). Since we have many matches between LP/MLS





and LP/OSIRIS we further binned data in the 5-degree and 10-degree latitude bins, respectively. We computed correlation coefficients between pairs of instruments in a similar manner for wide and/or 5/10-degree latitude bins.

For dense samplers like MLS and OSIRIS there are hundreds of matching profiles per day. To examine the stability of the LP ozone record and to study the ozone temporal evolution and global distribution, we constructed daily zonal mean (dzm)
time series by binning matching profiles for each day in 5- or 10-degree latitude bins for LP/MLS and LP/OSIRIS, respectively. From these dzm time series we further created monthly zonal mean (mzm) time series. Since we have more than five years of observations, we derived seasonal ozone cycles by averaging all mzm values for a specific month.

To assess the stability of the LP ozone record we first remove seasonal cycle independently from individual mzm time series and compute differences between deseasonalized anomalies. Then a linear regression is fitted to the differences between
deseasonalized anomalies. We deseasonalize anomalies to reduce persistence in the time series of residuals. This way we can assume that the residuals are random and normally distributed. Linear regression provides a simple way to estimate a linear drift. The standard error for the slope of the linear regression is estimated using:

$$\sigma_{slope}(z) = \frac{\sqrt{\frac{d^2(z)}{N-2}}}{\sqrt{\Sigma_{i=1}^{N}(p_i - \bar{p})^2}} , \qquad (4)$$

where $d^2(z) = \Sigma_{i=1}^{N}(\Delta_i(z) - a - bp_i)^2$ is the sum of the squared deviations between the seasonal anomaly differences $\Delta$
and linear regression fits $(a + bp)$ at altitude $z$, $N$ is the number of months, $p$ is time and $a$ and $b$ are regression coefficients. Assuming a normal distribution, the 95% confidence level of the slope is estimated as two times the standard error (Wilks, 2006).

## 4. Results

This section presents the evaluation of OMPS LP ozone retrievals in version 2.5. The section is divided into several
subsections. In section 4.1 we show results of the internal analysis and summarize our theoretically estimated systematic error for OMPS LP. Section 4.2 presents comparisons between the new version 2.5 and the previous version 2. Section 4.3 shows mean differences between OMPS LP version 2.5 and correlative satellite measurements. Section 4.4 is dedicated to analyses of global ozone patterns. Section 4.5 evaluates the stability of the version 2.5 OMPS LP record, and section 4.6 reviews the quality of the ancillary data.

### 4.1 Internal analysis of OMPS LP measurements

Limb scatter retrievals typically suffer from altitude registration and stray light errors. OMPS LP is no exception. In a first of its kind design, the LP CCD detector simultaneously measures solar radiances scattered from the atmospheric limb between 0 and 80 km and wavelengths from 290 to 1000 nm, significantly reducing the cost and improving the reliability of the instrument. This approach has its drawbacks related to internally scattered stray light and false vertical structures caused
by radiometric errors (Jaross et al., 2014).



The altitude registration methods and altitude adjustments applied to version 2.5 processing are described above in Section 2.4. The combined accuracy of the two altitude registration methods is about ±200 m. For LP a 200 m uncertainty in tangent height leads to 4-6% error in ozone at altitudes above 35 km (see Fig. S2 in the Supplement), and currently it is the largest source of known systematic errors in LP measurements (see Table 2). It is also important to ensure that there is no drift over

time in the sensor pointing. Any drift in altitude registration will inevitably translate into an artificial trend in the LP ozone dataset. This is why after version 2.5 processing was completed, our team checked the measured radiances and residuals using our altitude resolving methods. One of the methods (RSAS) has detected a 100-meter drift in sensor pointing over 5.2 years that is clearly seen in radiances from all 3 slits (see Fig. S3 in the Supplement). The drift in altitude registration seems to occur after mid 2016. Results from the ARRM method cannot undeniably confirm this drift. Our LP team is working to

resolve differences between the two methods and attribute the observed drift. The total error budget, summarized in Table 2, accounts for these uncertainties in accuracy and stability of the sensor pointing.

Radiometric errors also contribute to the systematic error in LP ozone retrievals. The OMPS LP ozone algorithm uses altitude normalized radiances such that any radiometric errors that are independent of altitude will cancel. Thus our main concern is radiometric errors that vary with altitude. These include stray light errors, small spectral shifts that change with

altitude, and small changes in the CCD detector efficiency in both vertical and spectral dimensions.

The initial analysis of LP ozone retrievals has revealed persistent artificial structures in LP ozone retrievals (~±3%). The downward propagating patterns observed in comparisons with MLS are also found in comparisons with other independent instruments (see Fig. 3 a-b). These structures repeat from orbit to orbit with remarkable precision. There is also evidence that these structures have changed over 5+ years of instrument operation, possibly as the result of altitude or spectral registration

drifts (see section 4.5 for details).

To map these ozone anomalies to a specific instrument problem we analyzed the Level 1 radiance measurements (Fig. 3 c). Because the original OMPS LP ozone retrieval algorithm has been simplified in the version 2.5 processing by reducing the number of wavelengths, the retrieved ozone now is more sensitive to Level 1 measurement errors at specific wavelengths and altitudes. Our analysis revealed subtle systematic errors (~±1-3%) in measured radiances (Fig. 3 c) not seen in calculated

radiances (Fig 3 d), which were computed with the GSLS radiative transfer model using the OMPS LP retrieved ozone profiles. These errors are most likely due to small variations in CCD pixel-to-pixel calibration in both the vertical and spectral dimensions. The geometry of LP observations changes over the orbit such that the same CCD pixel sees different altitudes along the orbit; this effect is more prominent in the southern hemisphere. This systematic error leads to artificial latitudinal structures in the LP ozone data.

Attributing and removing these instrumental artifacts is a high priority for the NASA OMPS LP Team. We are currently working on resolving these errors through calibration updates. Alternatively, if this issue cannot be resolved through calibration updates, we will search for an algorithmic solution to minimize the sensitivity of the LP retrievals to these subtle structures. But for now, we include these systematic errors in the total error budget (Table 2).





The goal of this study is to evaluate the OMPS LP ozone dataset, check our estimates of the systematic errors (described above), and isolate and characterize observed errors that cannot be fully explained by the known systematic errors. The attribution of observed ozone errors to a specific cause is a challenging process. The underlying instrument errors tend to produce complex patterns in ozone retrievals that vary with latitude and season.

Observed biases between the three LP slits demonstrates the complexity of the problem (see Fig. S4). Some fraction of these biases are due to remaining altitude registration offsets between the slits. The systematic radiometric errors differ from slit to slit creating artificial latitudinal structures in the ozone biases between the slits. The analysis of measured radiances also confirmed that stray light errors are much greater in the left and right slits compared to the center slit. For this reason, it was decided we would not release data from the left and right slits at this time. Only data from the center slit has been publically

released in this new version 2.5 processing and we focus below on analyses of LP data from the center slit only.

## 4.2 Comparisons between OMPS LP version 2 and version 2.5

All LP data starting from April 2012 have been processed with version 2.5, which now has more than 5 years of overlap with the previous version 2. In this section we provide comparisons between the two versions over the time period from April 1, 2012 up to May 1, 2017. Mean differences between the two versions as a function of latitude for UV and VIS ozone LP

retrievals are shown in Figures 4 and 5, respectively. Differences in the UV range have a clear latitudinal pattern with somewhat positive differences over the southern hemisphere and negative differences in the northern hemisphere, and a transition region over the southern subtropics. These differences are mostly driven by the intra-orbital, seasonally varying corrections in the sensor altitude registration implemented in version 2.5 (see Fig. 1). The intra-orbital corrections for the center slit vary by about 400 m between the south and north poles, with a positive adjustments of ~200-300 m in the

beginning of the orbit and negative correction -100 m at the end of the orbit. These altitude corrections lead to consistent, latitude-dependent biases between the two versions at all altitudes above ~30 km (see the top panel in Fig. S5 in the Supplement). The static altitude correction and two 100 m steps also contribute to the overall differences between the versions.

Since the new version uses only three UV pairs, the altitudes where the algorithm transitions from one wavelength pair to

another (37.5 km and 43.5 km) are now clearly seen in the differences. In addition, the increased sensitivity of the version 2.5 algorithm to the systematic errors in the measurements means that vertical oscillations and artificial latitudinal structures becomes more pronounced in the ozone retrievals. In general, the differences between the two versions in the UV range are smaller than 10% (and mostly within ±5%) and fully agree with our expectations based on the changes implemented in version 2.5 processing.

The largest changes from version 2 to version 2.5 are observed for the VIS retrievals with up to 60% differences in the lowermost stratosphere (see Fig. 5 and Fig. S5 in the Supplement). The aerosol correction module implemented in version 2.5 is responsible for smaller ozone values in the lowermost stratosphere and upper troposphere in the new version. Stratospheric aerosols attenuate solar light at the tangent point, and, if not properly accounted for, this effect leads to





overestimation of ozone in the satellite retrievals. Version 2 retrievals tended to systematically overestimate ozone in the lowermost stratosphere, because the aerosol correction module was turned off. The reduction of ozone in version 2.5 is expected to significantly improve the OMPS LP ozone product in the UTLS region compared to version 2 (see Fig. S6 in the Supplement). We observe an ozone increase in the middle stratosphere between 20-25 km, which is more pronounced in the

tropics and northern mid-latitudes. This increase is most likely related to the combined effects from altitude registration corrections (that decreases ozone above the peak and increases it below), aerosol and stray light corrections. The effect of the altitude correction is larger in the tropics (see Fig. S2 in the Supplement) and smaller in mid- and high-latitudes, thus it seems like a plausible explanation of the ozone changes in the tropics.

Two 100 m altitude corrections in April 2013 and September 2014 are mostly responsible for the relative drift between the

two versions (see Fig. 6). The intra orbital-and static correction also contribute slightly to the relative drift, modifying the amplitude at different latitudes. A 200-m change in altitude registration will create about 4-6% drift over 5 years in ozone above ~35 km. The observed relative drift between the two versions is about 1%/yr or 5% over 5 years, which is in agreement with the implemented corrections. However, above 43 km the drift is somewhat stronger, close to 2%/yr which might point to a possible drift in shorter UV channels or time-dependent changes in the stray light either at those altitudes or

at the normalization altitude (55 km).

## 4.3 Comparisons with independent satellite datasets

OMPS LP started operational measurements in April 2012, and currently LP has more than 5 years of overlap with several satellite missions: Aura MLS, Odin OSIRIS and ACE-FTS. We compared OMPS LP with MLS up to September 2017, with ACE-FTS up to August 2017 and with OSIRIS up to June 2017. The total numbers of matching profiles between pairs of

instruments in each wide latitude bin are shown in Fig. 7. Individual ozone retrievals for OMPS LP and ACE-FTS are illustrated in Fig. S7 in the Supplement along with the reported precisions. In many cases LP and ACE agree within the reported precisions, but in some cases the differences between two instruments are larger than reported random error bars. This is true for all considered pairs of instruments.

The mean biases for UV LP ozone retrievals between 29.5 and 53.5 km against correlative measurements are presented in

Fig. 8. The upper panel of Fig. 8 shows mean biases between LP and MLS as functions of latitude and altitude (5-degree latitude bins), and the middle panel demonstrates biases relative to OSIRIS (10-degree latitude bins). Lower panels in Fig. 8 show mean biases against all three instruments as functions of altitude in five wide latitude bins. The standard errors of the mean are smaller than the width of the lines. In the UV range biases are well within +/-10% (in many cases within ±5%) between 30 and 45 km, which satisfy the initial requirements for the LP retrievals (Rault and Loughman, 2013). Between 29

and 41 km LP tends to show larger values than OSIRIS and the relative LP bias is of order 4 -10%, while biases versus MLS and ACE-FTS in this altitude range are < ±5% and tend to oscillate around 0%. The standard deviations of differences, shown in Figure S8 in the Supplement, are ~5-10% and do not vary significantly with altitude up to 47 km. Above 47 km standard deviations tend to increase up to 10-12%.



Above 43 km we observe consistent negative biases against all considered instruments. The estimated systematic error for LP retrievals above 40 km is 7-9 % and ~5-7 % for MLS. The observed differences between LP and MLS are within the combined systematic error bars for these instruments. A systematic positive bias for ACE-FTS retrievals of ~10-20% in the upper stratosphere and mesosphere has been reported by Sheese et al. (2016). Our study also shows that ACE-FTS

overestimates ozone in the upper stratosphere compared to LP, however, biases between LP and ACE-FTS exceeds -20% above 50 km. in this region, the biases between OSIRIS and LP range between -8 and -15%. There is a common feature across all comparisons: biases are very close to 0% at 41 km and start to rapidly increase above.

The diurnal cycle (e.g. Parrish et al., 2014) can explain a small fraction of the observed differences among instruments in the upper stratosphere and mesosphere, where ozone rapidly decreases at the sunrise and relaxes back after the sunset driven by

photochemistry. This effect becomes prominent at altitudes above ~45 km, and day to night differences could rich up to 10% at 52 km. Because ACE-FTS is a solar occultation instrument and makes measurements during sunrise and sunset, and OSIRIS makes measurements at ~6 pm local time, they are expected to measure larger ozone concentration than the instruments that observe ozone during the daytime (Parrish et al., 2014). However, the diurnal cycle cannot fully explain the differences we found in our analysis.

Other apparent features in the spatial patterns of differences are persistent structures caused by the systematic errors in LP measurements. These artificial structures in OMPS LP data produce systematic errors in the LP measurements that become apparent when LP data are compared with other measurements. These errors are responsible for vertical oscillations in the LP retrievals and produce a pronounced pattern as a function of latitude (see discussion in Sec. 4.1).

It is important to note here that the observed patterns in differences for UV retrievals are not consistent with the patterns that

would be expected if there is an offset in altitude registration. The shift in the altitude registration would produce an error in ozone that has the same sign and magnitude at all altitudes above ~30-35 km (see Fig. S2 in the Supplement). Our comparisons do not reveal such patterns, which allows us to conclude that the LP altitude registration is within the reported ±200 m. The observed biases do not vary with latitude above 30 km (see Fig. S9 in the Supplement), confirming our implemented intra-orbital corrections.

The biases for the VIS LP retrievals are shown in Fig. 9. Biases are well within the required 10% above 18 km with the exception of the northern mid-latitudes (north off 45°N) between 21 and 30 km, where we observe systematic negative biases of order 10-15 %. These negative biases are caused by the unexpected thermal sensitivity of the instrument, which affects both pointing knowledge and wavelength registration. In addition to the thermal problem, sunlight comes closest to entering the sensor aperture as the instrument approaches the northern terminator. The first order corrections for the thermal

effect had been implemented in both versions – version 2 and version 2.5, but the residual effects are still seen in the ozone data.

In the tropical UTLS, the biases get larger. We observe large positive biases ~30% in the tropical lower stratosphere (17-20 km) against OSIRIS measurements. We also see positive biases against MLS but much smaller in magnitude (up to 6-12%). In the tropical upper troposphere LP has a negative bias by ~20-30% against all instruments.





In the mid-latitude lower stratosphere, LP tends to have a negative bias against all instruments with smaller biases against OSIRIS. Differences are typically smaller than 10%, except for the southern mid-latitudes (40S-60S) below 19 km, where we see large negative biases up to 15% relative to MLS and ACE-FTS. However, in version 2.5, the biases get smaller in the lower stratosphere than we observed in version 2 (see Fig. S6 in the Supplement), because of the explicit aerosol correction implemented in the new version.

We also computed differences for partial ozone columns in UV range between 31.5 and 52.5 km and VIS range between 12.5 km and 30.5 km against MLS and OSIRIS, shown in Fig. S10 in the Supplement. For UV partial columns we see agreement with MLS within 1%, but a positive bias of 2-3% against OSIRIS. For VIS partial columns we see good agreement with MLS in the tropics with biases close to 0 and positive biases of ~5% with OSIRIS. In the subtropics we see negative biases against both MLS and OSIRIS on the order of 3-7%.

In the altitude range between 29.5 km and 35.5 km VIS and UV retrievals from OMPS LP overlap. Our analysis show that there is a consistent bias between UV and VIS retrievals. Figure 10 shows mean differences between UV and VIS retrievals for several altitudes between 30 and 35 km. The differences range from -5 to 15 % with overall positive bias meaning that UV retrievals tend to overestimate ozone compared to VIS. Smaller differences are observed over the tropics, where they are mostly not significant within $1\sigma$ standard deviation. In the northern mid- and high latitudes there is a systematic bias of 5-15%, which is due to observed negative biases in VIS retrievals caused by the thermal effect. There is also a positive bias of ~15% in the high southern latitudes ~60°S. The main reason for the observed differences between UV and VIS retrievals is the inconsistency in calibrations between UV and VIS measured radiances. Despite large efforts invested in radiance calibrations the remaining uncertainties in the radiometric calibration and altitude and wavelength registration produce these offsets between UV and VIS ozone retrievals. We recommend using VIS retrievals up to 31 km. In the rest of this paper wherever we show profiles with an altitude range from 12 to 53 km, they are a combination of UV and VIS OMPS LP retrievals with UV only retrievals used between 31.5 km and 52.5 km, and VIS only between 12.5 and 30.5 km (no averaging or weighting).

We also computed correlation coefficients between matching OMPS LP ozone profiles and correlative measurements (see Fig. 11). We see strong correlation (>0.75) for the entire altitude range in the subtropics. The correlation with MLS seems to be stronger than with other instruments, probably due to greater number of matches and typically smaller temporal and spatial differences between OMPS and MLS measurements. It is interesting to note that the correlation drops in the northern and southern mid-latitudes around 25 km, which corresponds to altitude just above the peak in ozone number density. The correlation is smaller in the tropics, ranging from 0.4-0.8 above 18 km, and gets weaker in the tropical upper troposphere. The weaker correlation in the tropical stratosphere could be due to smaller natural ozone variability in this latitudinal zone. There is also a drop in correlation in the tropics around 27 km.





## 4.4 Global ozone variability

It is important to check how well OMPS LP can measure ozone variability and characterize vertical ozone distribution in different atmospheric regions most sensitive to changes in the stratospheric composition and dynamics. Figure 12 shows the mean ozone profiles and ozone variability relative to the mean as a function of latitude for OMPS LP, MLS and OSIRIS.

These metrics were calculated at a 5-degree latitude grid for OMPS LP and MLS and at a 10-degree grid for OSIRIS. Overall, good agreement can be seen between the three instruments in defining the global pattern of the ozone vertical distribution. The altitude of the maximum ozone concentration, vertical gradients and latitudinal patterns are very similar across the instruments. There is a discontinuity in OMPS LP data around 31 km due to differences between UV and VIS retrievals (described in the previous section). The patterns of the ozone variability indicate that in the tropical middle and

upper stratosphere ozone does not change much, and the typical variability there is less than 10%. The variability increases in the mid- and high- latitudes, where it also strongly depends on the season (see Fig. S11 in the Supplement). The greatest variability is observed in the lower stratosphere, where this quantity represents a combination of the real ozone variability and measurement noise. The percent variability differs the most from instrument to instrument in the tropical upper troposphere, indicating that instrumental effects play a dominant role in this part of the atmosphere, although the fact that

ozone abundances are smaller in this region plays a role as well.

To further check how well OMPS LP captures ozone variability, we derived seasonal cycles from dense sampling instruments: OMPS LP, Aura MLS and OSIRIS. Seasonal cycles were derived from monthly zonal mean time series, which were constrained from the matching ozone profiles. Figure 13 shows ozone seasonal cycles for 4 different altitude levels and three latitude bins. We narrowed the mid-latitude bins to 60°S-40°S and 40°N-60°N, respectively, because ozone seasonal

patterns change a lot across the 40-degree bins that we used to derive mean biases. For OMPS LP we show two ozone climatologies: one is sub-sampled to match with MLS measurements (grey lines in Fig. 13) and another with OSIRIS (black lines in Fig. 13). Calculation of seasonal patterns from OSIRIS data is complicated by the pattern of OSIRIS measurements, which leave us with several months of missing data. It was noted by Toohey et al (2013) that the ozone seasonal values, derived from different satellite measurements, could have significant biases (>10%) due to differences in the sampling of

observations. The leading cause of these differences is related to a nonuniform temporal sampling, but a nonuniform latitudinal/longitudinal sampling also contributes, especially at high latitudes (Toohey et al., 2013). Our results for OMPS LP also demonstrate significant differences in the ozone climatological values caused by different sampling. It is also important to note here that the seasonal values are expressed in percent from the corresponding overall means, and for the pair of OMPS LP and OSIRIS the mean values are calculated only from several months of the year due to the OSIRIS sampling,

while for the pair of OMPS LP and MLS means are computed from the entire year. OMPS LP seasonal cycle agrees remarkably well with the values derived from MLS and OSIRIS in terms of amplitude and phase with the exception of northern mid-latitudes (40°N-60°N) at 30 km, where the thermal sensitivity issue with OMPS LP has a strong seasonal component.





Figure 14 shows seasonal patterns derived from OMPS LP and MLS measurements independently for three latitude bins as function of month of the year and altitude. This figure once again demonstrates how well OMPS LP and MLS measurements agree in describing the ozone seasonal cycle. However, there are some individual layers where we see differences in the amplitude or phase of the seasonal cycle between LP and MLS. For instance, in mid-latitudes around 45-50 km we see that

MLS shows a slightly stronger amplitude of the seasonal cycle than OMPS LP. In the lower stratosphere vertical and horizontal (or temporal) gradients in the seasonal ozone cycle are very strong, which can lead to a shift in the phase of the seasonal cycle between instruments. These differences in seasonal cycle in the lower stratosphere could be due to differences in vertical resolution between the instruments. Though the instruments have very similar vertical resolutions, but due to the large ozone variability in the lower stratosphere, even small differences in the resolution can lead to significant biases.

Additional analysis is required to quantify and understand the remaining differences in the seasonal cycles, and this task is outside of the scope of this paper.

The residual seasonal biases between the instruments (remaining biases after removing the mean bias) are shown in Figs. S12 and S13 in the Supplement. The remaining seasonal biases typically do not exceed 5% and mostly are not statistically significant within $1\sigma$ standard deviation. The larger seasonal biases, which are also consistent between LP/MLS and

LP/OSIRIS, are observed at 30 km in the northern mid-latitudes (40°N-60°N), due to the thermal sensitivity issue with OMPS LP that tends to vary seasonally.

In the tropical stratosphere the Quasi Biennial Oscillation (QBO) is the leading cause of inter-annual ozone variability. In late 2015, an anomalous upward displacement of westerly winds started to develop at 20 hPa (Newman et al., 2016). It was accompanied by the development of easterlies in the 30-70 hPa layer. The anomalous westerlies disrupted the easterly phase

downward propagation at 10 hPa in late 2015 and early 2016. As a result trace gases in the tropical stratosphere responded to this disruption. Tweedy et al. (2017) demonstrated that during the 2016 boreal summer, total ozone was lower in the extratropics than during previous QBO cycles because of the change in the circulation caused by this disruption.

Figure 15 shows the ozone response to the disrupted QBO as measured by OMPS LP and Aura MLS. The two instruments demonstrate very similar patterns in the vertical distribution of tropical ozone and the temporal evolution of anomalies

driven by the equatorial QBO. These examples provide us with confidence that the calibration and algorithmic adjustments performed in version 2.5 do not affect OMPS LP ability to accurately and precisely derive ozone signals associated with natural variability.

## 4.5 Stability of the OMPS LP version 2.5 ozone record

Due to the relatively sparse pattern of ACE-FTS observations, it is very difficult to reliably estimate relative drift against

those data. Thus, we only estimated relative drifts against MLS and OSIRIS (see details in Sec. 3.4). Results are shown in Fig. 16. The positive drift in ozone between OMPS LP and MLS is on the order of 0.5-1.0 %/yr and is more pronounced at altitudes above 35 km. A smaller positive drift is seen against OSIRIS as well. Larger drifts are seen in the tropical UTLS. The vertical oscillations and latitudinal patterns in drift results (Fig. 16 upper and middle panels) are most likely caused by



the systematic errors in measured radiances described in Sec. 4.1. In the extratropics, drifts in the lower stratosphere below the ozone peak are mostly negative. Such patterns in the drift (with positive values above the ozone peak and negative values below) are consistent with the detected 100-meter drift in the LP altitude registration. These observed drifts in the LP ozone record are larger than the required stability for the LP sensor, which is 2% over 7 years (Rault and Loughman, 2013).

The time series of differences in UV ozone partial columns (30.5-52.5 km) shown in Fig. S14 in the Supplement also indicate that a drift started to develop in the middle of 2016. This is consistent with the results from the RSAS method (Fig. S3 in the Supplement), which also shows a drift beginning in the second part of 2016. The OMPS LP Team is working to resolve observed drifts in the altitude registration using a combination of two altitude resolving methods. Our goal is to reduce uncertainties in the altitude registration methods, especially for estimation of the sensor stability, to provide a climate

quality independent ozone record.

**4.6 Ancillary data**

Use of ancillary information for converting units and scale interpolation can potentially lead to additional sources of errors. Any systematic errors in ancillary data can potentially affect ozone comparisons. In this section we provide a brief evaluation of ancillary data that we use in this study.

To convert MLS ozone data we used pressure and temperature profiles from MERRA-2 FP reanalysis system (Gelaro et al., 2017). ACE-FTS retrieves atmospheric temperature and pressure along with the trace gases. Thus, we compared MERRA-2 and ACE-FTS temperature and pressure profiles (Fig. 17). Our results indicated that temperature biases are within ±3 K and are not statistically significant within 1σ standard deviation. The biases in temperature are somewhat larger near ~ 50 km, but these temperature biases are too small to explain large ozone biases above 45 km. We observe persistent biases of ~2-

2.8% in pressure that are only statistically significant in the tropics and do not vary much with altitude. Observed differences in pressure can be translated into ~150 m relative offset in the altitude registration between OMPS LP and ACE-FTS sensors, which is smaller than ±200 m combined error for the LP altitude registration. Boone et al, (2005) reports that ACE-FTS pointing accuracy is ±100 m.

In addition, we compared MERRA-2 temperature and pressure profiles with those derived from ECMWF reanalysis (Dee et

al., 2011). Results for the 40.5 km altitude level are shown in Fig. S15 in the Supplement and we found overall good agreement in terms of accuracy, precision and stability. These results tell us that the ancillary data that we use for analysis do not have large systematic errors that can significantly alter our ozone comparisons.

**5. Conclusions**

In summer 2017 all OMPS LP measurements starting from April 2012 have been processed with the new version 2.5

algorithm. Key changes implemented in this new version include three types of corrections for the sensor pointing, the new cloud detecting algorithm, stray light corrections for the VIS radiances, explicit aerosol corrections and a reduction of the number of wavelengths used in the retrievals.





To verify the implemented calibrations, algorithmic changes and sensor pointing corrections, we compared the LP version 2.5 ozone retrievals against Aura MLS, ACE-FTS and Odin OSIRIS satellite observations. Our analysis shows that OMPS LP retrievals accurately characterize the vertical ozone distribution in different atmospheric regions which are most sensitive to changes in the stratospheric composition and dynamics. Specifically, in this study we show that LP measurements agree

well with MLS and OSIRIS in reproducing ozone natural variability associated with the seasonal cycle and Quasi Biennial Oscillations in terms of amplitude, phase, and vertical structure.

Our analysis indicates that between 18 and 42 km the mean differences between LP and correlative measurements are well within required ±10%, with the exception of the northern high latitudes where we observe larger biases between 20 and 32 km due to a remaining thermal sensitivity issue. In the upper stratosphere and lower mesosphere (>43 km) OMPS LP tends

to have a negative bias against all considered instruments. We also find larger biases in the lower stratosphere and upper troposphere, especially in the tropics where biases could be up to -30%. However, we see significant improvements in version 2.5 compared to version 2 due to the implemented aerosol correction. Larger than 10% negative differences against MLS and ACE-FTS are also observed in the southern mid-latitudes (20°S-60°S) below 18 km.

It is important to note, that our comparisons confirm that the absolute LP altitude registration is well within ±200 m

combined uncertainty for the LP altitude registration methods. Our results also confirm intra-orbital altitude corrections, which helped to remove the latitudinal dependence in biases that we saw in version 2. We found a small positive drift of ~0.5%/yr against MLS and OSIRIS that is more pronounced at altitudes above 35 km. Such a pattern is consistent with the 100-meter drift (over 5 years) in sensor pointing detected by one of our altitude resolving methods.

Most of the observed uncertainties in OMPS LP ozone retrievals are related to remaining errors in instrument calibration and

altitude registration. The attribution of observed errors to a specific cause is a challenging process, as errors in ozone produced by various causes tend to interfere and produce complex patterns. Both external (comparisons with independent observations) and internal (analysis of LP radiances) validation results are critical for evaluating LP altitude registration and calibrations. We expect this type of work to continue throughout the life of the instrument.





**Acknowledgements:** Authors would like to thank Didier Rault, who developed the original ozone retrieval algorithm for the OMPS LP. We also would like to thank Robert Loughman for developing and maintaining the radiative transfer code for the OMPS LP and Ghassan Taha for developing the empirical stray light corrections for VIS radiances. We are grateful to all members of the OMPS Team for their work on supporting the OMPS mission. This work was funded under the National
Aeronautics and Space Administration (NASA) project on Development of Ozone Vertical Profiles from OMPS LP (WSB 437949). Work performed at the Jet Propulsion Laboratory, California Institute of Technology, was performed under contract with the National Aeronautics and Space Administration (NASA). Authors would like to acknowledge the Canadian Space Agency for their continued support of the Odin OSIRIS and ACE-FTS missions. This paper contains some materials that were published earlier this year in the short, overview article in the NOAA GSICS Quarterly Newsletters (Kramarova et
al., 2017, doi: 10.7289/V5R78CFR). This paper includes some materials that were first presented at the 9th Atmospheric Limb Symposium (Saskatoon, Canada, June 2017) and at the 2017 STAR Annual JPSS meeting (College Park, MD, USA, August 2017).

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



**Table 1.** Summary of key changes implemented in versions 2 and 2.5 relative to the base algorithm described in (Rault and Loughman, 2013).

| Key changes | Version 2 | Version 2.5 |
|---|---|---|
| **Cloud height detection** | (Rault, 2005) | New algorithm to better discriminate between clouds and aerosol (Chen et al., 2016) |
| **Altitude Registration** | **Static corrections** of 0.58/1.18/1.75 km for the left/center/right slits, correspondently | **Static corrections** of 1.12/1.37/1.52 km for the left/center/right slits, respectively; **+0.1 km adjustments for all 3 slits** on April 25, 2013 and on September 5, 2014; **Intra-orbital and seasonal TH** adjustments of ~0.3-0.4 km. (Moy et al., 2017) |
| **Stray Light Corrections** | Prelaunch SL corrections; Corrections for unexpected thermal sensitivity in NH (Jaross et al.,2014) | Empirical corrections for VIS (similar to those described in (Taha et al., 2008)) |
| **Wavelengths Selection** | **UV:** 289-325 nm paired with 353 nm (about 43 UV pairs); **VIS:** 549-633 nm combined with the reference wavelengths at 510 nm and 673 nm to form ~17 VIS triplets | **UV:** 302 nm, 312 nm and 322 nm paired with 353 nm (3 UV pairs); **VIS:** 600 nm combined with 510 nm and 675 nm to form a single VIS triplet |
| **Radiance Normalization Altitude** | 65 km for UV and 45 km for VIS | **UV:** 55 km **VIS:** 40 km |
| **Aerosol correction** | No explicit aerosol correction | Use aerosol extinction coefficient profiles retrieved from LP measurements for same event (Loughman et al., 2017) |
| **Vertical Smoothing** | $2^{nd}$ order Twomey-Tikhanov regularization term | Define a priori covariance matrices $S_a$ assuming 25% ozone variability above 20 km, 50% ozone variability below 16 km and 5 km inter-level correlation |
| **Measurement error** | **UV:** 1% **VIS:** 1% | **UV:** 1% **VIS:** 0.5% |



Table 2. Known systematic errors in OMPS LP ozone retrievals.

| Altitude (km) | Error in O3 (%) due to ±200 m error in altitude registration | Drift in O3 (%/yr) due to drift in altitude registration (~100 m/5.2 yrs) | Errors in O3 (%) due to systematic errors in the measured radiances |
|---|---|---|---|
| <15km | ~±5-10 | ~0.5-1.0 | ±3 |
| 20 km | ~±10 | ~1.0 | ±3 |
| 25 km | ~±0 | ~0 | ±3 |
| 30 km | ~±2 | ~0.2 | ±3 |
| 35 km | ~±5 | ~0.5 | ±3 |
| 40 km | ~±5 | ~0.5 | ±3 |
| 45 km | ~±5 | ~0.5 | ±3 |
| 50 km | ~±5 | ~0.5 | ±3 |





**Figure 1. Intra orbital and seasonally varying corrections in the OMPS LP altitude registration derived from the ARRM method (see text for the details) for each LP slit. The corrections are expressed in meters and shown as functions of event number along the orbit (X-axis) and day of the year (Y-axis). Event number along the orbit does not directly correspond to a specific latitude, because OMPS LP acquires solar scattered measurements and the latitude where LP makes its first measurement on orbit varies seasonally with the solar zenith angle. Our analysis of measured radiances strongly suggests that the altitude registration error depends on the event number (or the time that the instrument is exposed to solar light) rather than the geographic latitude.**





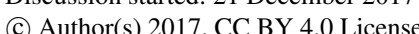

**Figure 2. Systematic and random errors for satellite ozone retrievals in the mid-latitudes (a) and tropics (b). Different colors correspond to various satellite instruments. Dashed lines show random measurement errors (retrieval precisions) for all considered satellite instruments. Solid lines show known, reported systematic errors for OMPS LP version 2.5 and MLS version 4 ozone retrievals in percent from the mean.**





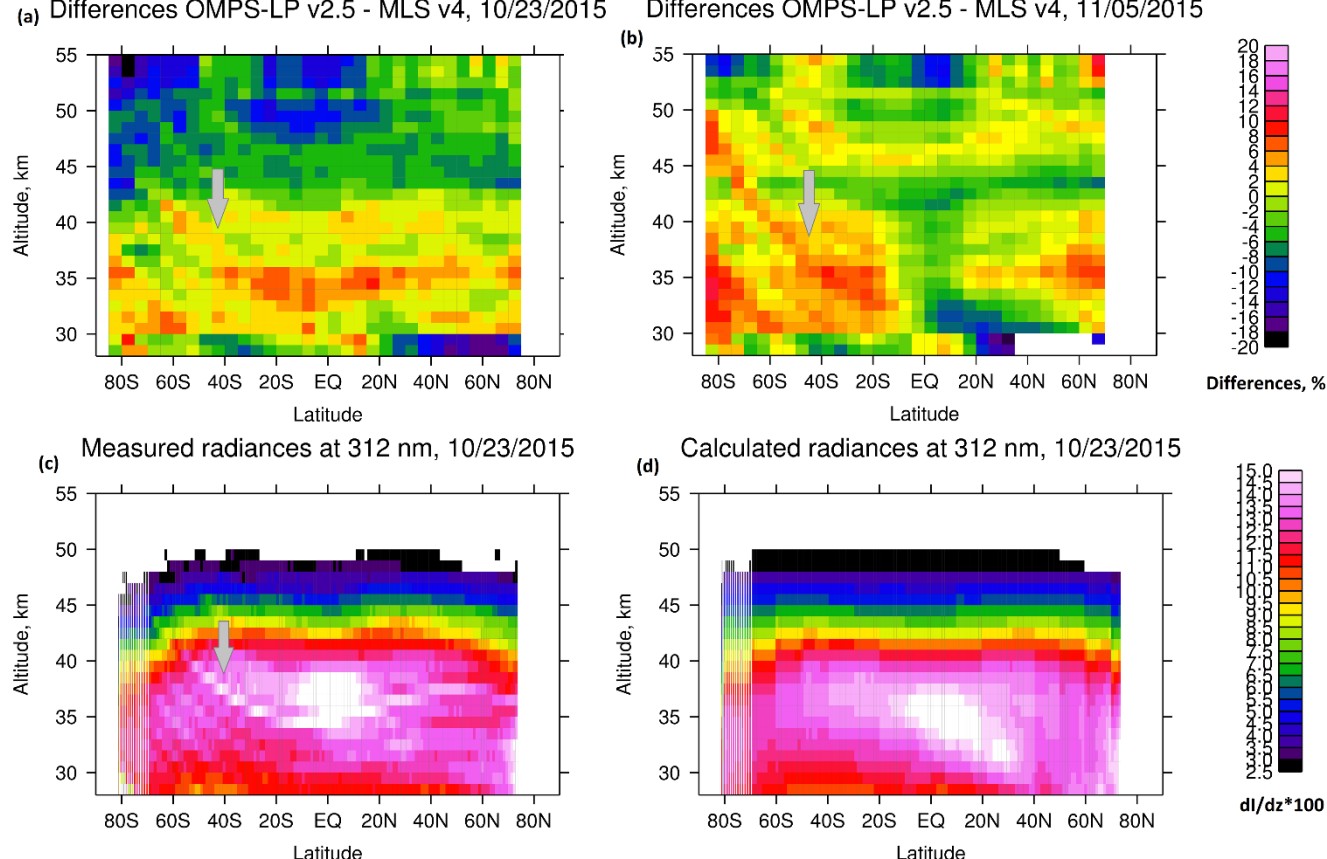

**Figure 3. Upper panels (a-b) show mean differences (%) between LP and MLS ozone profiles on October 23, 2015 and two weeks later on November 5, 2015. The grey arrows show one of the artificial structures found in LP ozone retrievals that repeat from orbit to orbit with remarkable precision. Lower panels (c-d) show results of the internal analysis of measured (c) and calculated (d) radiances at 312 nm on October 23, 2015. The subtle structures in measured radiances (not seen in calculated radiances) mimic structures observed in LP ozone retrievals.**





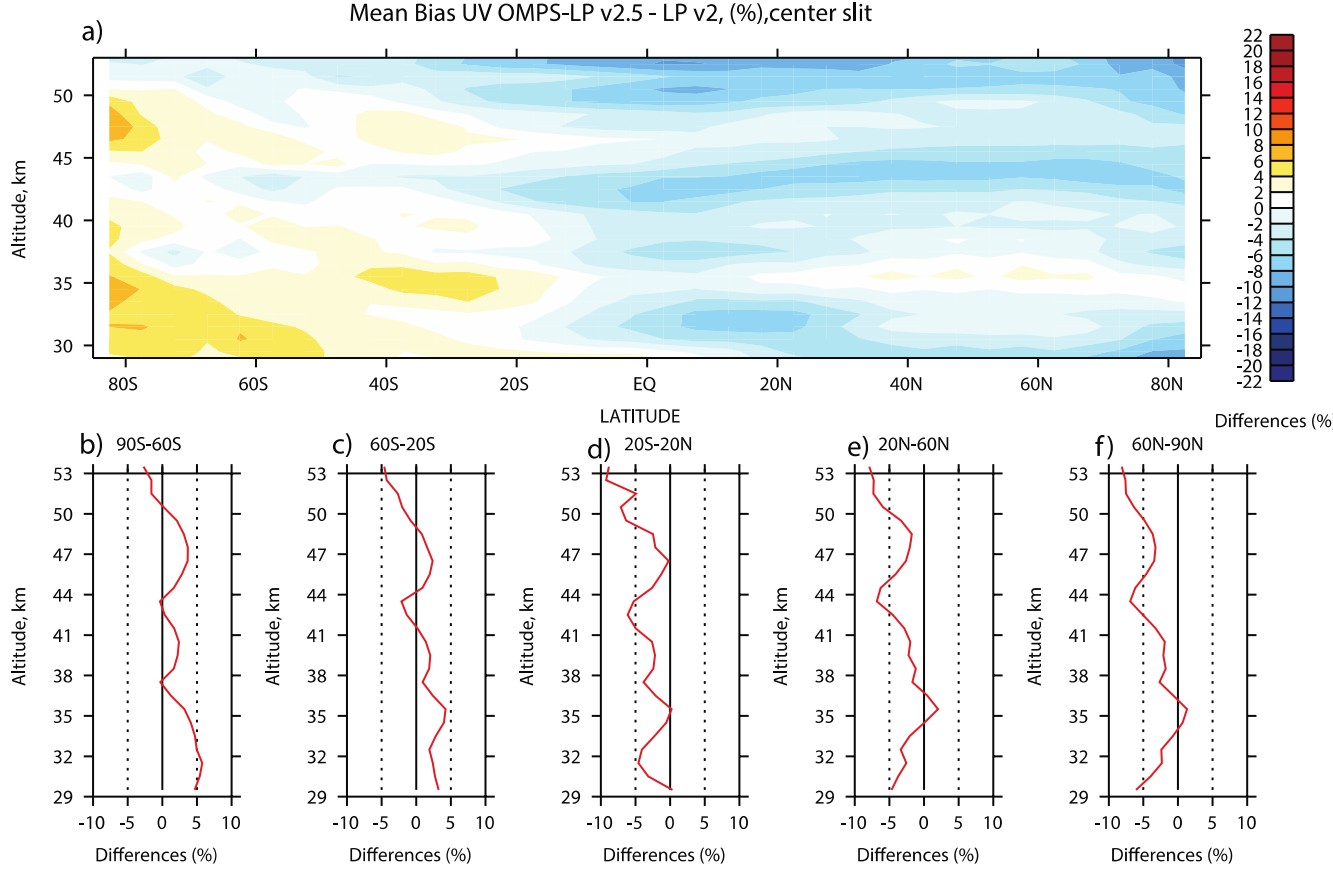

**Figure 4: Mean biases in the UV ozone retrievals (29.5km-53.5 km) between the two OMPS LP versions: version 2.5 and version 2. Upper panel (a) shows the bias as a function of latitude and height (on 5° latitude grid). The lower panels (b-f) show vertical profiles of the differences in five wide latitude bins.**





**Figure 5. Same as figure 5 only for the VIS ozone retrievals in the vertical range between 12.5 km and 35.5 km.**







**Figure 6. Relative drift between OMPS LP version 2.5 and OMPS LP version 2. Upper panel (a) shows the drift as a function of latitude and height (on 5° latitude grid). The lower panels (b-f) show vertical structures of the drift in 5 wide latitude bins. Horizontal error bars show 1σ errors for the linear fit. Red lines indicate drifts for the UV ozone retrievals and blue lines for VIS.**





**Figure 7. Total number of matching profiles for each pair of instruments in five wide latitude bins. Red bars show number of matching profiles between Aura MLS and LP, blue bars for Odin OSIRIS and LP, and green bars for ACE-FTS and LP. Please, note that the Y-axis show number of measurements in logarithmic scale.**



**Figure 8. Mean biases for the UV ozone retrievals from OMPS LP version 2.5 against correlative satellite measurements. Top panel (a) shows the bias with Aura MLS version 4 as a function of latitude (on $5^0$ latitude grid) and height. Middle panel (b) shows the bias with Odin OSIRIS version 5.10 (on $10^0$ latitude grid). Lower panels (c-g)**
5  **illustrate vertical structures of the mean biases against Aura MLS (red), Odin OSIRIS (blue) and ACE-FTS (green) in five wide latitude bins. The standard error of the mean is smaller than the width of the lines.**



**Figure 9. Same as Figure 8 only for the VIS OMPS LP retrievals.**

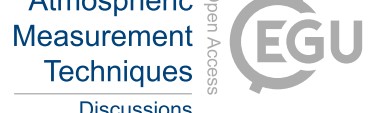



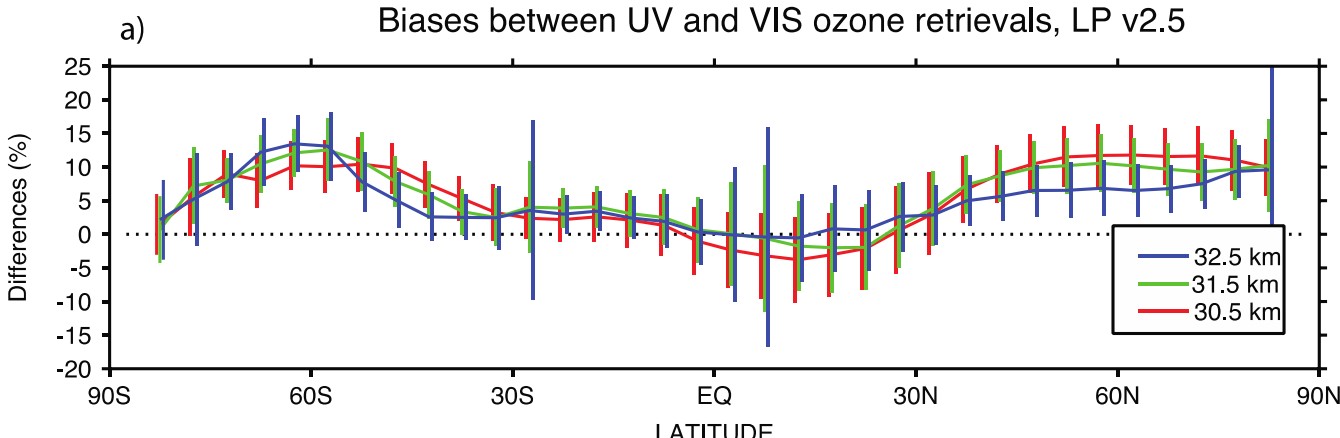

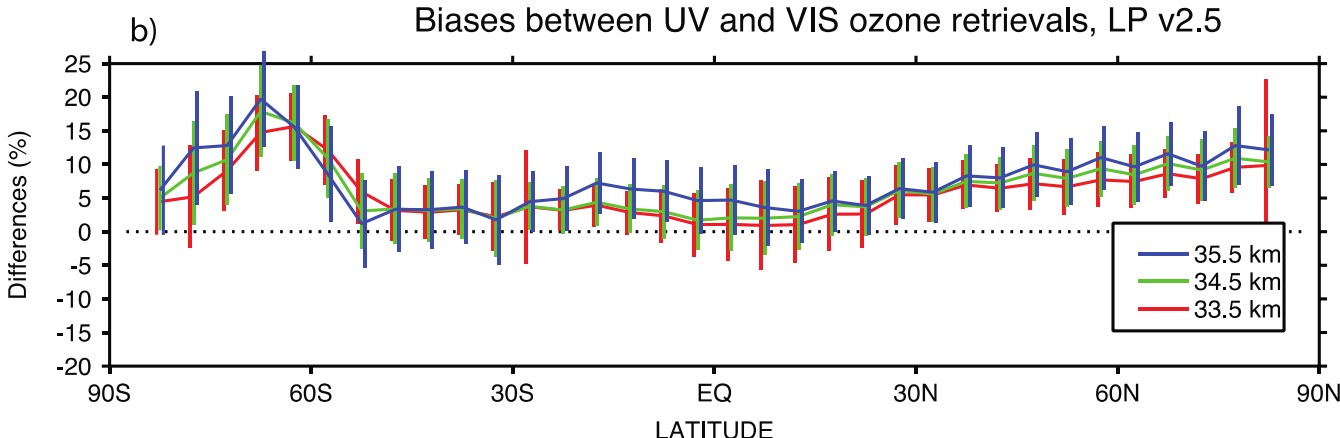

**Figure 10. Mean biases between the UV and VIS OMPS LP ozone retrievals in version 2.5 as functions of latitude. Upper panel (a) shows differences at 30.5, 31.5 and 32.5 km, while lower panel (b) presents differences at 33.5, 34.5 and 35.5 km. Error bars indicate 1σ standard deviations of the differences.**



**Figure 11. Correlation coefficients between OMPS LP version 2.5 ozone profiles and correlative satellite measurements. Upper panel (a) shows the correlation between OMPS LP v2.5 and Aura MLS as a function of latitude and altitude (on 5º latitude grid). Middle panel (b) shows the correlation between LP and OSIRIS (on 10º latitude grid). The lower panels (c-g) show vertical structures of the correlation in five wide latitude bins.**





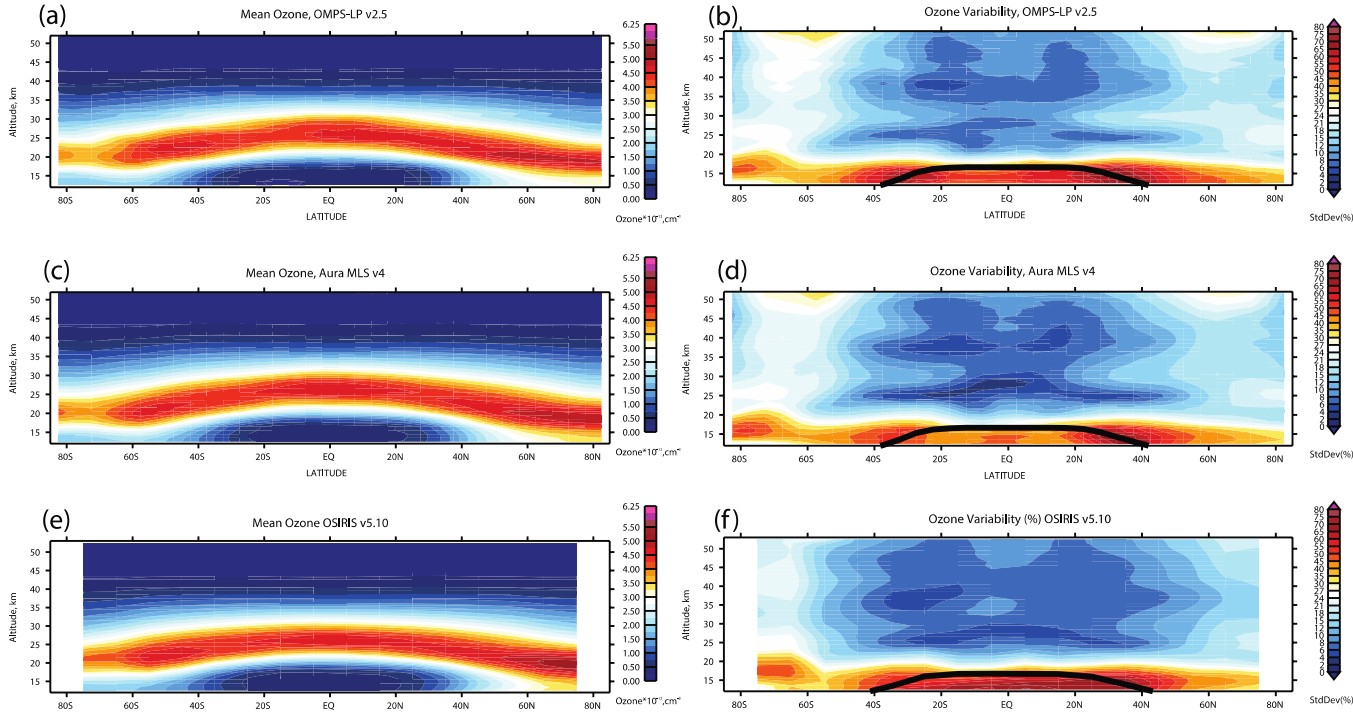

**Figure 12. Mean ozone values (left column) in cm$^{-3}$ and ozone variability (right column) in % as functions of latitude and altitude for three satellite instruments: OMPS LP (upper panels (a-b)), Aura MLS (middle panels (c-d)) and OSIRIS (lower panels (e-f)). Data for OMPS LP and MLS are shown on 5º latitude grid, while OSIRIS values are shown on 10º latitude grid. The ozone variability is calculated as 1-σ standard deviations from all individual ozone retrievals relative to the instrument means (shown in the left column).**



**Figure 13. Seasonal ozone cycle (in % from the instrumental mean) for OMPS LP, MLS and OSIRIS in three wide latitude bins: 40° S - 60° S (left column), 20° S - 20° N (center column), and 40° N - 60° N (right column). The seasonal cycle is shown at several altitude levels: 20.5 km (a-c); 30.5 km (d-f); 40.5 km (g-i); and 50.5 km (j-l). OMPS LP seasonal values are calculated from both sub-samples: from matches with MLS (grey lines) and matches with OSIRIS (black). MLS seasonal values are shown in red and OSIRIS in blue. The standard errors of the mean are typically smaller than the width of the lines. The error bars indicate 1σ standard deviations.**





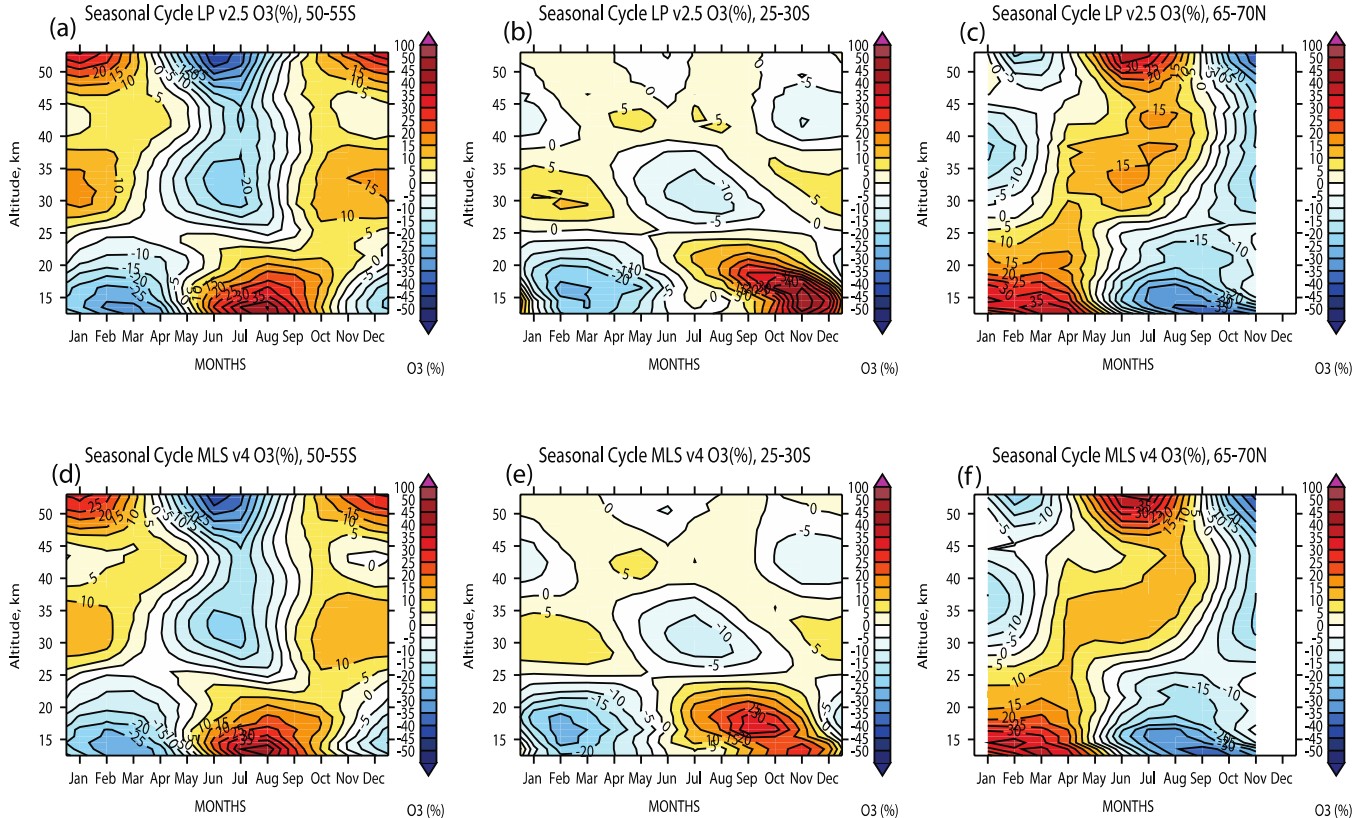

**Figure 14.** Seasonal ozone cycles derived from OMPS LP (upper row) and Aura MLS (lower row) as functions of month and altitude for three different 5° latitude bins: (a and d) for 50° S - 55° S, (b and e) for 25° S - 30° S, and (c and f) for 65° N - 70° N.

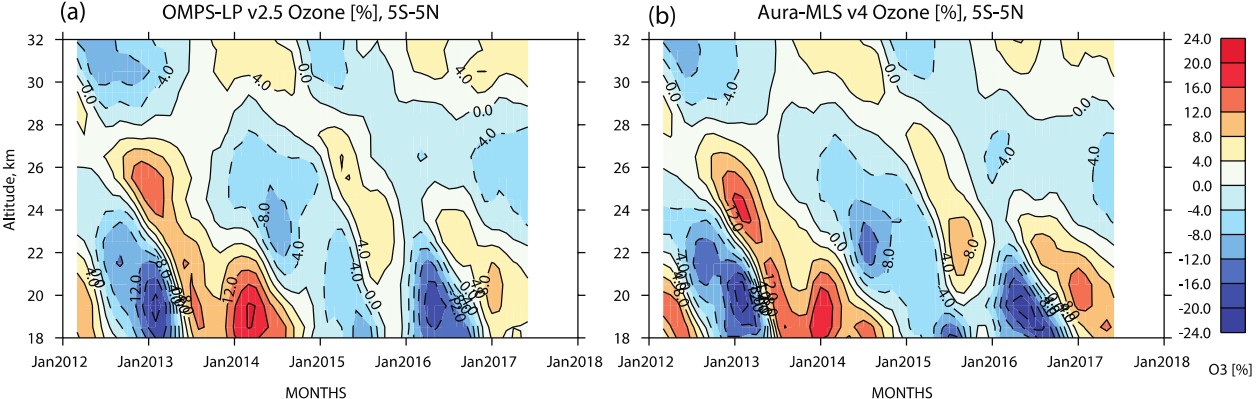

**Figure 15.** Deseasonalized ozone anomalies over the equatorial latitudes (5° S – 5° N) for OMPS LP (a) and Aura MLS (b). The Quasi-Biennial Oscillations (QBO) drives ozone distribution in the tropics. The anomalous disruption of equatorial zonal wind in late 2015 and 2016 led to changes in the tropical ozone distribution. These figures show that the two instruments show very similar responses in ozone to the anomalous QBO event.






**Figure 16. Relative drift for OMPS LP version 2.5 against Aura MLS and Odin OSIRIS. Upper panel (a) shows the relative drift between OMPS LP and MLS as a function of latitude and altitude (on 5° latitude grid). The center panel (b) illustrates relative drift between OMPS LP and OSIRIS (on 10° latitude grid). The lower panels (c-g) show vertical structures of the drift in 5 wide latitude bins against MLS (red lines) and OSIRIS (blue lines). Error bars show 1σ standard deviations for the linear fit.**




**Figure 17. Mean differences between MERRA-2 and ACE-FTS pressure (a-e) and temperature (f-j) profiles in five wide latitude bins. Biases for pressure are expressed in %, while temperature biases are shown in degrees [K].**
5   **Horizontal error bars show 1 σ standard deviations of the corresponding differences.**