# Peer review of "Validation of ozone profile retrievals derived from the OMPS LP version 2.5 algorithm against correlative satellite measurements"

_Atmospheric Measurement Techniques, 2017_

## Referee Comment (RC1) · Anonymous Referee #1 · 15 Jan 2018

General comments:

This paper deals with a new algorithm version for the OPMS LP data. The paper is well written, concise and contains interesting and important observations for users of OMPS LP data. I would like to recommend its publication in AMT. I have the following comments and questions.

Specific comments:

1. p.4, line 18: During past few years there has been activity to harmonize the ozone cross sections used in ground-based and satellite remote sensing instruments (see Oprhal et al., Journal of Molecular Spectroscopy, 327:105– 121, September 2016).

[Figure]

What is the attitude of the OPMS algorithm team towards this activity? 2. p.4, line 23: Normalizing by the high altitude radiance: If I understand right, you must do the same operation in the radiative transfer model i.e., dividing by the upper altitude model radiance. Please, provide some information how to calculate the reference upper level radiance. Whar do you assume about the atmospheric state? 3. p.4, line 29: Wavelength pairs and triplets: Are you doing flat field correction before pairing? 4. p.5, line 1: Why do you perform the ozone retrieval separately in UV and VIS wavelengths? There is only unique ozone field, so the results should agree within error limits inside some altitude interval, If not, something is wrong. Please, comment? 5. p5, line 20: You are moving from the Tikhonov smoothing to a full a priori covariance matrix. This adds an active a piori to the ozone density and is, therefore, at least in principle a stronger constraint for the solution (but this depends on used parameters). Please add reasons for this decision. If you compare your earlier Tikhonov and the new a priori regularisation, how do ozone profiles differ? 6. p.5, line 20: A priori information is a part of your solution. How are you following the amount of a priori information in your product and do you apply some maximum limit for the a priori contamination? 7. p.5, line 28: The discussion of the SNR is somehow confusing to me. What is measured now and what was assumed earlier? 8. p5, line 28: How do you measure SNR for unique measurements? Every situation is new. 9. p5, line 28: If the measured SNRs do not agree with the calculated SNR, is something wrong in the error propagation or in the estimation of the instrumental noise or what? 10. p 5, line 6: If I understand right, when you detect a cloud below your LOS you are still retrieving ozone above the cloud top. But how are you handling the cloud contribution to radiance in your radiative transfer model? 11. p5, line 23: The new aerosol mode: Are the artificial structures now removed? How do your aerosol profiles compare with aerosol profiles from other satellite missions? 12. p12, line 13: You are saying that altitude independent factors cancel out when using normalized radiances. In my mind only strictly multiplicative altitude independent factors cancel out. For example, stray light residuals and other additive factors are not cancelled. Also if the factors are inside spectral integral (point spread)

and/or integration time integral, they are not cancelled. Do you agree? 13. p16, line 18: You explain the observed UV-VIS differences by remaining radiometric calibration differences. But using normalised radiances and wavelength pairs or triplets should reduce the calibration sensitivity. Why these tricks do not work? 14. Sec 4.2: I wonder if you have calculated Chi2-values for the old and new version. Is the fitting quality improved? 15. Regarding the difficulties related to the instrument: What improvements for a future similar instrument can now be foreseen?

---

## Referee Comment (RC2) · Anonymous Referee #2 · 2 Mar 2018

This paper is clearly presented and well organized to describe a newly upgraded retrieval algorithm for OMPS LP observations. The subject of the paper is appropriate to AMT. Below are a few comments concerning clarifications / extensions for consideration in the final publication in AMT.

[1] Have the impacts of algorithm updates on the data throughput/yields been estimated? [2] For the comparisons among MLS, OSIRIS, ACE, have the contribution of the accuracy differences of spectroscopic parameter data across microwave-infrared-vis-UV ozone bands taken into account? The spectroscopic differences could be one of the observable sources that contribute to relative bias among data sets. Its quantifi-

cation could help in bias corrections of data products. [3] This paper has been focusing on the comparisons of OMPS LP central slit measurements with reference data sets. Could you consider to include discussions on the evaluation of the quality of OMPS LP retrievals using the measurements from OMPS LP left and right slits? or any possible approaches (e.g., via data assimilation system(s) + reference data sets) of estimating the quality of those retrievals?

―――――――――――――――――――――――

---

## Author Comment (AC1) · 29 Mar 2018

Referee: "General comments: This paper deals with a new algorithm version for the OPMS LP data. The paper is well written, concise and contains interesting and important observations for users of OMPS LP data. I would like to recommend its publication in AMT. I have the following comments and questions."

Authors: Authors would like to thank the referee for reviewing the manuscript and providing constructive comments. Authors' responses are below.

Specific comments: Referee: 1. p.4, line 18: During past few years there has been ac-

[Figure]

tivity to harmonize the ozone cross sections used in ground-based and satellite remote sensing instruments (see Orphal et al., Journal of Molecular Spectroscopy, 327:105–121, September 2016). What is the attitude of the OPMS algorithm team towards this activity?

Authors: The OMPS team is closely monitoring the work and new developments in the field of the ozone spectroscopy. We are committed to use the most accurate ozone cross sections. For the future processing, our plan is to update ozone cross sections in the UV and VIS ranges. However, the impact of the change in ozone cross sections on LP ozone profile retrievals is expected to be small (∼1-2%).

Referee: 2. p.4, line 23: Normalizing by the high altitude radiance: If I understand right, you must do the same operation in the radiative transfer model i.e., dividing by the upper altitude model radiance. Please, provide some information how to calculate the reference upper level radiance. What do you assume about the atmospheric state?

Authors: We normalize measured and calculated radiances at 55 km and 40 km for UV and VIS, respectively. We use ozone a priori profiles [McPeters and Labow, 2012] to calculate initial radiances at 55 km. For the following iteration steps, we adjust ozone climatology at upper altitudes above 50 km by applying a scaling factor derived based on the retrieved ozone values at 50 km to ensure a smooth transition between retrieved ozone profile and climatology. Since July 2016, in the forward processing algorithm, we removed the scaling factor and use the climatological profile above 50 km. We do not see large changes in the ozone retrievals due to this scaling. We included the following explanation into the text: "The ozone climatological profiles are used to simulate radiances at the normalization altitudes. From the beginning of the OMPS mission and until June 2017, the climatological profiles were scaled at each iteration based on the ozone values retrieved at the previous step at the level 5 km below the normalization altitude. Since July 2017 in the forward processing, we do not apply scaling and use the original climatological values to simulate radiances. Our analysis revealed very little differences in the ozone retrievals due to the climatology scaling."

Referee: 3. p.4, line 29: Wavelength pairs and triplets: Are you doing flat field correction before pairing?

Authors: The algorithm uses calibrated, sun-normalized radiances for all retrievals. The effect of this step is very small, but it is aimed to reduce the magnitude of the calibration errors.

Referee: 4. p.5, line 1: Why do you perform the ozone retrieval separately in UV and VIS wavelengths? There is only unique ozone field, so the results should agree within error limits inside some altitude interval, If not, something is wrong. Please, comment?

Authors: The Signal to Noise Ratio (SNR) for UV radiance measurements is higher than for VIS. Therefore if one would use UV and VIS radiances together, the algorithm would weight UV radiances more than VIS, even at altitudes where UV radiances loose useful information. This is the main reason for producing UV and VIS retrievals independently. Ideally, UV and VIS retrievals should match over the altitude range where they overlap. However, our results clearly show that there is a bias between UV and VIS retrievals. We believe that the remaining uncertainties in the instrumental calibrations between UV and VIS is the main reason for the observed biases (see our response on Q13 below). However, other factors like differences in the quality of the ozone cross sections between UV and VIS can contribute as well. Instead of blending UV and VIS data together and hiding a problem, the OMPS team chose to release data from UV and VIS ranges and continue investigating the causes of these biases.

Referee: 5. p5, line 20: You are moving from the Tikhonov smoothing to a full a priori covariance matrix. This adds an active a piori to the ozone density and is, therefore, at least in principle a stronger constraint for the solution (but this depends on used parameters). Please add reasons for this decision. If you compare your earlier Tikhonov and the new a priori regularization, how do ozone profiles differ?

Authors: The two methods are served the same purpose - to stabilize ozone profile retrievals and vertically smooth them. The weighting coefficients for the Tikhanov's

regularization are usually selected to match a desirable vertical resolution. In a case of the LP, these weights were selected for ozone and aerosol retrievals independently to match the sensor FOV [Rault & Loughman, 2013]. These weights varied with altitude and also depended on a number of spectral channels used in the retrievals. The OMPS team decided to use Rodgers' approach and set up the a priori covariance matrices with non-zero off-diagonal elements to stabilize the LP retrievals. We believe that this method is more transparent and have a clear physical meaning. The diagonal elements of the a priori covariance matrices were selected to closely match the observed ozone variability (based on McPeters&Labow climatology). For the off-diagonal elements, we assumed the exponential decline in the inter-level correlation within 5 km. Rodgers called the measurement and a priori covariance matrices as the retrieval 'tuning parameters'. We chose those parameters based on analysis of the ozone variability and LP measurements (see answer on Q7 below) to achieve a consistent vertical resolution that is close to the sensor FOV.

Referee: 6. p.5, line 20: A priori information is a part of your solution. How are you following the amount of a priori information in your product and do you apply some maximum limit for the a priori contamination?

Authors: Due to a nature of the limb technique, the measured radiances provide sufficient content of information and the retrieval algorithm does not rely on the a priori. The careful selection of the wavelengths and the cut-off altitudes (altitudes below/above which that wavelength has a very weak sensitivity to ozone) further ensure that LP measurement vector has sufficient information to retrieve ozone profile independently from the a priori information.

Referee: 7-9. p.5, line 28: The discussion of the SNR is somehow confusing to me. What is measured now and what was assumed earlier? How do you measure SNR for unique measurements? Every situation is new. If the measured SNRs do not agree with the calculated SNR, is something wrong in the error propagation or in the estimation of the instrumental noise or what?

Authors: Thank you for pointing this. We modified the text to clearly distinguish a difference between the SNR, which is reported in the Level 1 product as a detector noise, and the measurement uncertainty, which is something introduced in Level 2 processing for the purpose of constructing the measurement covariance matrix. The SNR, or detector noise, is a calculated quantity, which is found to closely match the detector noise estimated through other means. This quantity did not change in the new processing. We changed the measurement uncertainty in the retrieval algorithm. In the previous version we assumed that the measurement uncertainties are equal to the inversed SNR. However, the analysis of the LP measurements revealed that the SNR underestimate the actual measurement uncertainties, therefore we increased the measurement uncertainties in v2.5 retrieval algorithm. We modify the text: "The measurement noise covariance matrix S_Ïţ in version 1 was set as a diagonal matrix with each diagonal element being associated with the Signal to Noise Ratio (SNR) for the corresponding wavelength at a given altitude. The SNR is reported for every single measurement, and it varies from 320 for 290 nm at 60 km to 1200 for 600 nm at 15 km. The SNR is a calculated quantity that is aimed to accurately characterize the sensor's detector noise. The analysis of the random errors in LP measurements showed that the SNR significantly underestimates the actual measurement noise (Jaross et al., 2014) that varies in a range between 0.5% and 1%. It is important to clearly distinguish a difference between the SNR and the measurement uncertainty, which is introduced in the inverse model for the purpose of constraining ozone retrievals. For version 2, the measurement noise was assumed to be ∼1% for both UV and VIS retrievals. In the current version 2.5, the measurement noise is prescribed to be 1% in UV spectral range and 0.5% in VIS."

Referee: 10. p 5, line 6: If I understand right, when you detect a cloud below your LOS you are still retrieving ozone above the cloud top. But how are you handling the cloud contribution to radiance in your radiative transfer model?

Authors: This is correct. The LP algorithm retrieves ozone profiles above the cloud top.

[Figure]

An effective surface reflectance is computed that represents a weighting average of the surface and cloud reflection, considering any clouds as being present at the terrain height. Radiances at 675 nm are used to estimate the reflectivity. We clarified that in the text: "If a cloud is detected (see Sec. 2.3 below), an effective surface reflection is computed using measurements at 675 nm to represent a weighting average of the surface and cloud reflection, considering any clouds as being present at the terrain height."

Referee: 11. p5, line 23: The new aerosol mode: Are the artificial structures now removed? How do your aerosol profiles compare with aerosol profiles from other satellite missions?

Authors: The new aerosol correction scheme works better, and we do not observe any artificial structures in the ozone retrievals due to the aerosol correction. The LP v1 aerosol profiles had been compared with OSIRIS and CALIPSO, and on average LP aerosol agrees within 20-25% [Loughman et al., 2017]. We added this note in the text: "The LP v1 aerosol extinction profiles had been compared with independent observations from OSIRIS and CALIPSO, and on average LP aerosol extinction agrees within 20-25% (Loughman et al., 2017)."

Referee: 12. p12, line 13: You are saying that altitude independent factors cancel out when using normalized radiances. In my mind only strictly multiplicative altitude independent factors cancel out. For example, stray light residuals and other additive factors are not cancelled. Also if the factors are inside spectral integral (point spread) and/or integration time integral, they are not cancelled. Do you agree?

Authors: Yes this is correct. Sun-normalization cancels radiometric calibration errors and most bandpass errors. Altitude normalization also cancels these errors to the extent they are altitude-independent. Stray light is not a radiometric calibration error. It is a form of non-linearity, which the various normalizations definitely do not address.

Referee: 13. p16, line 18: You explain the observed UV-VIS differences by remaining

radiometric calibration differences. But using normalised radiances and wavelength pairs or triplets should reduce the calibration sensitivity. Why these tricks do not work?

Authors: Altitude normalization and wavelength pairing indeed reduce many radiometric errors (see answer on Q12 above). The radiances in UV and VIS ranges are measured by the OMPS LP in two different instrumental modes. In order to capture measurements from all altitudes and to avoid saturation of CCD pixels, the measurements are done through large and small aperture and at two different integration times (Jaross at al, 2013). The measurements in UV range come from the large aperture, while VIS radiances from the small aperture. Analysis of in-flight measurements (Jaross at al., 2013) revealed that radiances for the same wavelength and altitude systematically differ by several percent between large and small aperture, most likely due to stray light, relative errors in the tangent height between the two apertures, instrumental thermal sensitivity, and radiance gridding errors. As a result we see differences between UV and VIS ozone retrievals.

Referee: 14. Sec 4.2: I wonder if you have calculated Chi2-values for the old and new version. Is the fitting quality improved?

Authors: We did not make an attempt to fit a statistical model to the LP ozone data record to derive ozone trends and/or analyze ozone variability.

Referee: 15. Regarding the difficulties related to the instrument: What improvements for a future similar instrument can now be foreseen?

Authors: The next LP sensor will be on board of the JPSS-2 satellite. The design of the OMPS LP instrument for JPSS-2 has incorporated several lessons-learned from SNPP OMPS. First and foremost, a structural modification of the inlet baffle should limit the optical distortion caused when the SNPP external baffle is heated by the sun. This structural change will remove the thermal sensitivity effect that affected LP VIS measurements, therefore improving retrievals in the NH lower stratosphere. This baffle is also extended to better reject out-of-field stray light (Earth shine). Another structural

change is the installation of curtains between the slits that will prevent sunlight entering the right slit and to contaminate center and left slits. The large aperture will be filtered to allow only UV light, thus better rejecting VIS stray light. The small aperture will be filtered to attenuate VIS signals and enhance NIR signals, thus improving stray light performance in NIR. A new coating has been used on the Limb CCD to improve measurements in the NIR.
* * *

---

## Author Comment (AC2) · 29 Mar 2018

Referee: This paper is clearly presented and well organized to describe a newly upgraded retrieval algorithm for OMPS LP observations. The subject of the paper is appropriate to AMT. Below are a few comments concerning clarifications / extensions for consideration in the final publication in AMT.

Authors appreciate the referee's comments and provide point-to-point responses below.

[1] Referee: Have the impacts of algorithm updates on the data throughput/yields been

[Figure]

estimated?

Authors: In the new version 2.5 we implemented two types of changes: calibration updates (including altitude corrections) and algorithmic changes. In Sec. 4.2 we compare v2.5 retrievals against the previous version 2. Where it is possible we attribute observed changes in ozone to a specific calibration/algorithmic update.

[2] Referee: For the comparisons among MLS, OSIRIS, ACE, have the contribution of the accuracy differences of spectroscopic parameter data across microwave-infrared-vis-UV ozone bands taken into account? The spectroscopic differences could be one of the observable sources that contribute to relative bias among data sets. Its quantification could help in bias corrections of data products.

Authors: We agree with the referee that the differences in the accuracy of spectroscopic data between UV/VIS, microwave and infrared spectral ranges can be responsible for a fraction of the observed biases among the instruments. However, in the presented study we did not attempt to account for these uncertainties.

[3] Referee: This paper has been focusing on the comparisons of OMPS LP central slit measurements with reference data sets. Could you consider to include discussions on the evaluation of the quality of OMPS LP retrievals using the measurements from OMPS LP left and right slits? or any possible approaches (e.g., via data assimilation system(s) + reference data sets) of estimating the quality of those retrievals

Authors: Internally, we analyzed the ozone retrievals from left and right slits, compared them with the data from the center slit and against independent instruments. We see consistent biases between the three slits that can not be explained by the geophysical ozone variability. We also see larger number of outliers in the retrievals from the side slits. The analysis of measured radiances revealed larger SL and calibration problems in the left and right slits (altitude and spectral shifts). The separation between 3 slits is about 250 km, and the stratospheric ozone does not change much across 250 km, except for the polar winter and spring conditions (ozone hole). In most cases, data from

the side slits don't bring additional information about the ozone distribution. Considering significantly larger measurement errors in the side slits, the OMPS Team decided to release only ozone retrievals from the center slit at this time. However, we continue efforts to characterize and reduce errors in the side slits.
* * *

---

## Author Response (AR2)

**Response to the editor's comments:**

**Editor:** "Thank you for addressing the referee's comments. Your paper can be accepted in AMT after minor corrections:

p. 4, line 32: "revealed very little differences" should be quantified, please provide a range of % differences or < x%.

General: please provide links and DOIs, if available, to the data sets used."

**Authors:**

We would like to thank the editor for thoughtful suggestions. Our responses are below:

P4, line 32, we changed the text: "Our analysis revealed very little differences (~1-2%) in the ozone retrievals at levels 5-7 km below the normalization altitude due to the climatology scaling".

We added a section "Data availability" on p. 20 just before the "Acknowledgements" and provided links to the datasets used in our study:

*"Data availability.* The OMPS LP version 2.5 ozone profile dataset (doi:10.5067/X1Q9VA07QDS7) used in this study is publically available at https://disc.gsfc.nasa.gov/datasets/OMPS_NPP_LP_L2_O3_DAILY_2/summary. Aura MLS data are available at https://mls.jpl.nasa.gov. Odin OSIRIS ozone profiles can be found at http://odin-osiris.usask.ca/?q=node/280. The ACE-FTS Level 2 data can be obtained via the ACE website (registration required): http://www.ace.uwaterloo.ca."

The revised manuscript is below with all changes marked up.

[revised manuscript text omitted]